



# Censored rainfall modelling for estimation of fine–scale extremes

David Cross[1], Christian Onof[1], Hugo Winter[2], Pietro Bernardara[2]

[1]Department of Civil and Environmental Engineering, Imperial College London, London SW7 2AZ, UK

[2]EDF Energy R&D UK Centre, London SW1E 5JL, UK

*Correspondence to*: David Cross (david.cross12@imperial.ac.uk)

**Abstract.** Reliable estimation of rainfall extremes is essential for drainage system design, flood mitigation and risk quantification. However, traditional techniques lack physical realism and extrapolation can be highly uncertain. In a warming climate, the moisture holding capacity of the atmosphere is greater which increases the potential for short duration high intensity storm events. In this study, we improve the physical basis for short duration extreme rainfall estimation by simulating

the heavy portion of the rainfall record mechanistically using the Bartlett-Lewis rectangular pulse model. Mechanistic rainfall models have had a tendency to underestimate rainfall extremes at fine temporal scales. Despite this, the simple process representation of rectangular pulse models is appealing in the context of extreme rainfall estimation because it is emulates the known phenomenology of rainfall generation. A censored approach to Bartlett-Lewis model calibration is proposed and performed for single site rainfall from two gauges in the UK and Germany. Extreme rainfall estimation is performed for each

gauge at the 5, 15 and 60 minute resolutions, and considerations for censor selection discussed.

## 1. Introduction

With growing evidence that the frequency and intensity of short duration rainfall extremes are increasing with climate change (Stocker et al. 2013, Westra et al. 2014, Kendon et al. 2014), the need for reliable extreme value estimation techniques is becoming more pressing. Extreme rainfall estimation is required for numerous applications in diverse disciplines ranging from

engineering and hydrology to agriculture, ecology and insurance. It facilitates the planning, design and operation of key municipal infrastructure such as drainage and flood defences, as well as scenario analysis for climate impact assessment, and hazard risk modelling. Extremes are usually estimated using frequency techniques and intensity duration frequency curves. However, these methods are highly dependent on the observed rainfall record which may not be characteristic of the extreme behaviour.

In this study we improve the physical basis of short duration extreme rainfall estimation by simulating the heavy portion of the observed rainfall time-series. Traditional approaches to extreme value estimation rely on sampling extremes from the observed record. However, rainfall observations present various problems for the practitioner. They are often not available at the location of interest, they are typically short in duration, and they may not be available at the temporal scale appropriate for



the intended use. These difficulties, together with the necessity to obtain perturbed time-series representative of future rainfall,

have motivated the development of stochastic rainfall generators since the earliest such statistical models developed by Gabriel & Neumann (1962). The reader is referred to Waymire & Gupta (1981), Wilks & Wilby (1999) and Srikanthan & McMahon (2001) for detailed reviews of early developments in rainfall simulation.

The principle of rainfall simulation is to replicate statistical properties of the observed record such that multiple realisations of statistically identical rainfall may be synthesized (Richardson 1981). Various methods of simulation exist, and there have been

several attempts in the literature to categorize the different approaches. Aside from dynamic methods used in numerical weather prediction models, Cox & Isham (1994) suggest that statistical simulation methods may be broadly categorized as either purely statistical or stochastic, while Onof et al. (2000) further categorize stochastic methods into either multi-scaling or mechanistic. The latter of these differ from other statistical approaches because rainfall synthesis follows a simplified representation of the physical rainfall generating mechanism. Through the clustering of rain cells in storms, the unobserved

continuous-time rainfall is constructed by superposition, enabling the synthetic rainfall hyetograph to be aggregated to whatever scale is desired (Kaczmarska et al. 2014). Because of this simplified process representation, mechanistic model parameters have physical meaning which makes this class of model particularly appealing in the context of extreme value estimation.

When no likelihood function can be formulated (Rodriguez-Iturbe et al. 1988, Chandler 1997), mechanistic models are

typically calibrated using a generalised method of moments (Wheater et al. 2007a) with key summary statistics at a range of temporal scales such as the mean, variance, autocorrelation and proportion of dry periods. Performance is assessed on the ability of the models to reproduce statistics not used in calibration including central moments and extremes. Since their inception in the late 1980s by Rodriguez-Iturbe, Cox & Isham (1987, 1988), numerous studies have demonstrated the ability of these models to satisfactorily reproduce observed summary statistics [see Cowpertwait et al. (1996), Verhoest, Troch & De

Troch (1997), Cameron, Beven & Tawn (2000a, 2000b), Kaczmarska, Isham & Onof (2014), Wasko & Sharma (2017) and Onof et al. (2000) for a review]. However, these studies have also shown that mechanistic models tend to underestimate rainfall extremes at the hourly and sub-hourly scales which limits their usefulness [see Verhoest et al. (2010) and references therein]. We hypothesize that stochastic mechanistic pulse-based models may be poor at estimating fine–scale extremes because the training data, and calibration method, are dominated by low intensity observations. Mechanistic stochastic models are fitted to

the whole rainfall hyetograph, including zeroes, aggregated to a range of temporal scales. Typically, the range of scales used varies from hourly to daily, although implicit in most studies is the assumption that scales required in simulation should be within the range of scales used in calibration. Hence, if the intention of the model is to simulate 15 minute rainfall the training data should include 15 minute observations. As the temporal resolution of rainfall data becomes finer, the distribution of rainfall amounts becomes more positively skewed. Primarily, this is because of the increased proportion of dry periods, but

also the higher proportion of low intensity events characteristic of fine–scale rainfall. Because the calibration method uses





central moments to fit model parameters, the greater skewness at finer temporal scales makes it difficult to obtain a good fit to extremes at these scales.

In addition to the dominance of low observations, the estimation of fine–scale extremes may be further undermined by operation and sampling errors. This is particularly true of tipping bucket gauges where measurement precision at fine temporal scales is limited to the bucket volume, typically 0.2 or 0.5 mm. Fine–scale rainfall is highly intermittent (starting and stopping with high frequency), yet a tipping bucket gauge can only make a recording when the bucket is full. The limitations of tipping bucket measurements at fine temporal scales have long been understood (Goldhirsh et al. 1992, Nystuen et al. 1996, Yu et al. 1997), although the first formal estimation of sampling error was performed by Habib, Krajewski & Kruger (2001). In this study, the authors investigate the ability of tipping bucket gauges to capture the temporal variability of fine–scale rainfall at 1, 5 and 15 minute scales using tipping bucket measurements simulated from high resolution optical rain gauge observations. The authors show that for the lowest rainfall intensities (< 5 mm/h) the mean relative error of the tipping bucket gauge at the 5 minute resolution is + 3.5%, with corresponding standard deviation just under + 30% for a bucket volume of 0.254 mm. Larger errors are obtained for the 1 minute resolution. They also show that increasing the bucket volume to 0.5 mm significantly increases the spread of the sampling error for low observations at the 5 minute resolution. The observed record comprised mainly convective storm events which are typical for Iowa in the US where the data were collected, although the error estimates are significant and present compelling evidence of the impact of sampling error on fine–scale low intensity rainfall observations.

Significant effort has been made since the late 1980s to improve the performance of mechanistic rainfall models through structural developments, with substantial focus on the improved representation of fine–scale extremes (see Sect. 2 for a review). Despite this, little progress has been achieved. To test our hypothesis, a simple approach is proposed in which low observations for fine–scale data are censored from the models in calibration. This focusses model fitting on the heavier portion of the rainfall record at fine temporal scales, and reduces rainfall intensity at coarser scales. The aim is to investigate if existing mechanistic models can be used as simulators of fine–scale storm events by changing the data and not the model, thereby reducing the impact of low observations and sampling error on fine–scale extreme rainfall estimation.

The choice of models is limited to those within the Bartlett-Lewis family of models which conform to the original concept of rectangular pulses developed by Rodriguez-Iturbe, Cox & Isham (1987). Preference is given to the most parsimonious model variants on the basis that having fewer parameters improves parameter identifiability and reduces uncertainty. The Neyman-Scott family of models is excluded on the understanding that the clustering mechanisms of both model types perform equally well (Wheater et al. 2007a), and there is no evidence that randomisation of the Neyman-Scott model (Entekhabi et al. 1989) has any advantage over its Bartlett-Lewis counterpart.

In Sect. 2, we outline the main mechanistic model developments for improved representation of extremes. The Censored modelling approach for the estimation of fine–scale extremes is described in Sect. 3. Model structure and selection is explained

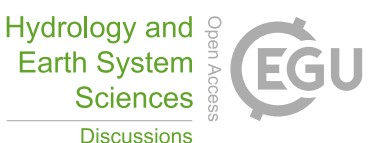

in Sect. 4, and the data and fitting methodology are presented in Sect. 5. Results are given in Sect. 6 together with validation

analysis. Discussion on censor selection and the results are given in Sects. 7 and 8. Section 9 outlines our main conclusions

and thoughts for future research.

## 2.    Mechanistic model developments

Attempts to improve the estimation of fine–scale extremes for point (single-site) rainfall using mechanistic models have

focused on changing the model structure. Several authors have cited significant improvement (Cowpertwait 1994, Cameron et

al. 2000b, Evin and Favre 2008), although increased parameterization and limited verification with real data have meant that

most changes have not been widely adopted. An early criticism of the original mechanistic models presented by Rodriguez-

Iturbe, Cox & Isham (1987) is that the exponential distribution applied to rainfall intensities is light-tailed. This choice is

consistent with the observation that rainfall amounts, which in the model are obtained through the superposition of such cells,

are approximately Gamma distributed (Katz 1977, Stern and Coe 1984).

On the basis that the Gamma distribution gives more flexibility in generating rain cell intensities, Onof & Wheater (1994b)

reformulate the modified (random η) Bartlett-Lewis (MBL) model (Rodriguez-Iturbe et al. 1987) with the Gamma distribution

to improve the estimation of extremes. Despite the good fit to hourly extremes cited by Onof & Wheater (1994b), subsequent

studies have continued to show underestimation at hourly and sub-hourly scales (Verhoest et al. 1997, Cameron et al. 2000a,

Kaczmarska et al. 2014).

In an extension of this approach, Cameron, Beven & Tawn (2000b) replace the exponential distribution in the MBL model

with the Generalized Pareto Distribution (GPD) for rain cells over a high threshold. Depending on the value of the shape

parameter ($\xi$), the GPD converges to one of three forms: upper-bounded ($\xi < 0$), exponential ($\xi = 0$) and Pareto ($\xi > 0$). In the

last case, we have a distribution with a heavier tail than the exponential or the gamma. Because the GPD is a model for

threshold exceedance, the authors specify a threshold below which the MBL model with exponential intensity distribution is

used to simulate rain cell depth, and above which the Pareto distribution is used. This is justified on the assumption that the

rain cell depth may be of either high or low intensity.

The authors present a calibration strategy in which they first fit the MBL model with exponential cell depths to the whole

rainfall record using the method of moments from Onof & Wheater (1994b). Generalised Likelihood Uncertainty Estimation

(Beven and Binley 1992) is then used to find behavioural parameterizations of the Pareto scale and shape parameters for rain

cell depths over the threshold – the location parameter being fixed at the threshold value. The central assumption of this model

is that the Pareto scale and shape parameters for cells depths over the threshold will have "minimal impact on the standard

statistics of the simulated continuous rainfall time-series" (Cameron et al. 2000b, p.206). The validity of this assumption is

disputed by Wheater et al. (2007a) who argue that the MBL model should be fitted to rainfall coincident with rain cells below

the threshold, but point out that this is "impossible since cell intensities are not observed" (Wheater et al. 2007a, p.16).



The model framework of Cameron, Beven & Tawn (2000b) differs from that of the MBL Gamma model of Onof & Wheater (1994a) and is essentially the nesting of two models. The authors present significant improvement in the estimation of hourly extremes and show good agreement with Generalized Extreme Value (GEV) estimates. However, because the underlying process of continuous-time rainfall is unobserved, the authors are forced to implement a calibration strategy which limits the impact on standard rainfall statistics – an approach which is undesirable (Wheater et al. 2007a). Furthermore, the framework appears to be an analogue of the N-cell rectangular pulse model structure initially developed by Cowpertwait (1994) for the Neyman-Scott model, and later incorporated into the Bartlett-Lewis models by Wheater et al. (2007a). Regardless of their relative performance, the large number of parameters required for these models is undesirable on the basis that more parameters reduces parameter identifiability and increases parameter uncertainty.

In an earlier study, Cowpertwait (1994) differentiates between light and heavy rain cells in a modified version of the original (fixed η) Neyman-Scott Rectangular Pulse (NSRP) model (Rodriguez-Iturbe et al. 1987) by allowing rain cell intensity and duration to be drawn from more than one pair of exponential distributions. The new model termed the Generalised NSRP model (GNSRP) leads to a significant increase in parameterization over the original NSRP model, although the author presents an intelligent way to simplify calibration by relating model parameters to harmonic signals. While improvement is achieved in the fit to hourly extremes, the performance of the model in replicating other important statistics is not presented, in particular autocorrelation and the proportion of dry periods. Both of these properties are addressed by Rodriguez-Iturbe, Cox & Isham (1987, 1988) for the Bartlett-Lewis model with the inclusion of a "high frequency jitter" and randomisation of the rain cell duration parameter η. Entekhabi, Rodriguez-Iturbe & Eagleson (1989) present a randomised version of the Neyman-Scott model with significant improvement in the fit to dry periods. However, because no analytical expression was available for the proportion of dry periods this statistic was not used in model fitting, and other model parameters were not allowed to vary from storm to storm with randomisation. Consequently, while the MBL and the GNSRP models each allow rain cell intensity and duration to be drawn from more than one pair of distributions, the MBL structure is preferred because it has fewer parameters.

In a later study, Cowpertwait (1998) hypothesised that including higher-order statistics in the fitting routine for mechanistic rainfall models would give a better fit to the tail of the empirical distribution for rainfall amounts. Focussing on the original (fixed η) NSRP model, analytical equations for skewness of the aggregated rainfall depth are presented and used in fitting the models. Empirical analysis showed that including skewness in the fitting statistics improved the estimation of Gumbel distribution parameters from simulated maxima when compared with parameters obtained from observed annual maxima.

A criticism of the rectangular pulse model structure by Evin & Favre (2008) is that it assumes independence between rain cell intensity and duration. Following previous attempts to link the two variables [Kakou (1997), De Michele & Salvadori (2003), Kim & Kavvas (2006)], Evin & Favre (2008) present a new NSRP model in which the dependence between rain cell depth and duration is explicitly modelled using a selection of copulas. While the authors are not primarily motivated to improve the





estimation of rainfall extremes, good estimation of fine–scale extremes is achieved. However, the manner in which the results are presented makes interpretation and comparison with other studies difficult. In the first instance, the extreme performance of all models is almost entirely indistinguishable indicating that no overall improvement is achieved. Secondly, monthly annual extremes are presented at hourly and daily scales but without clearly stating which month in the year. Despite this, it is likely

that monthly extremes will have less variance than those taken from the whole year, and hence model performance is likely to be better. On the basis of the results presented, it is not clear that explicitly modelling dependence between rain cell depth and duration with copulas offers any discernable benefit over the original model structure.

Theoretically, copulas offer an attractive framework for modelling the dependence structure between rainfall intensity and duration. However, the obvious mechanism for building copula dependence into mechanistic rainfall models is at the rain cell

level as per Evin & Favre (2008). This approach draws upon the intuition that, just as for the rainfall amounts of storm events, rain cell amounts may be correlated with their duration. Such intuition follows earlier studies into the dependence structure between rainfall intensity and duration (Bacchi et al. 1994, Kurothe et al. 1997) – although as stated by Vandenberghe et al. (2011, p.14) *"it is not very clear in which way this modelled dependence at cell level alters the dependence between the duration and mean intensity of the total storm"*.

In recent years, renewed focus on estimating rainfall extremes at hourly and sub-hourly scales has led to the development of a new type of mechanistic rainfall model based on instantaneous pulses (Cowpertwait et al. 2007, Kaczmarska 2011). In this model structure, rectangular pulses are replaced with a point process of instantaneous pulses with depth X and zero duration, the summation of pulses giving the aggregated time rainfall intensity. Considered initially to offer a more suitable representation of rainfall at sub-hourly scales than rectangular pulses, Kaczmarska, Isham & Onof (2014) found that the best

performing Bartlett-Lewis Instantaneous Pulse (BLIP) model effectively generated rectangular pulses when depth X was kept constant, and cell duration $\eta$ was randomized. Because of the very large number of pulses generated within cells, the authors noted that this model structure imposes the *"most extreme form of dependence"* Kaczmarska, Isham & Onof (2014, p.1977). Consequently, the authors developed a new rectangular pulse model in which both $\eta$ and $\mu_x$ are randomized (BLRPR$_X$) which was found to perform equally as well as the randomised version of the BLIP model but without additional parameterization.

**3.    Censored modelling for fine–scale extremes**

Despite the model improvements outlined in Sect. 2, there is an on-going tendency for stochastic mechanistic models to underestimate extremes at hourly and sub-hourly scales, requiring the practitioner to employ additional methods for better extreme value performance including disaggregation (Koutsoyiannis and Onof 2000, Koutsoyiannis and Onof 2001, Onof et al. 2005, Onof and Arnbjerg-Nielsen 2009, Kossieris et al. 2013). We propose a censored approach to mechanistic rainfall

modelling for improved estimation fine–scale extremes by focussing model fitting on the heavy portion of the rainfall time-series. The aim of this research is to investigate if mechanistic models can be used as simulators of fine–scale design storm



events to reduce the impact of low observations on the estimation of fine–scale extremes. In this approach, rainfall below a low censor is set to zero and rainfall over the censor is reduced by the censor amount. The effect is to generate a time-series of heavy rainfall based on the observed record in which the proportion of dry periods is increased, and rainfall amounts are

reduced.

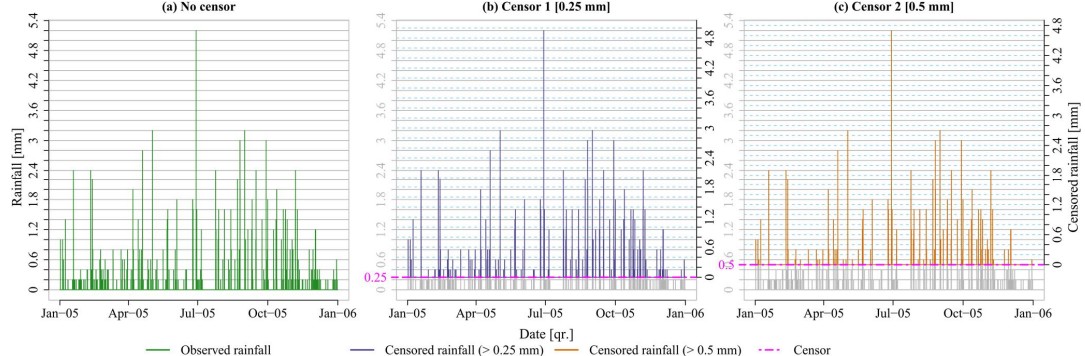

**Figure 1 Example censoring applied to 15 minute rainfall data at Atherstone in 2005. Arbitrary censors of 0.25 mm and 0.5 mm are applied to demonstrate the effect of censoring on the rainfall record.**

Figure 1 shows two arbitrary censors applied to 15 minute data at Atherstone in 2005 (refer to Sect. 5.1 for a description of the data). Panel (a) shows the uncensored rainfall, and the two right panels (b and c) the change in rainfall with increasing censors. The reduced rainfall amounts are shown on the secondary y axes. It can be seen from these plots that the minimum

recorded rainfall is 0.2 mm which corresponds to the tip volume of the tipping bucket rain gauge. Compared with higher rainfall amounts this volume is recorded with very high frequency throughout the year at the 15 minute resolution.

Censored rainfall synthesis is a method for estimating sub-hourly to hourly extremes. Because observations below the censor are omitted from model fitting, censored model parameters are scale dependent and can only be used to simulate storm profiles above the censor at the same scale as the training data. It is the ability to simulate the heavy portion of storm profiles which

enables extreme rainfall estimation. The basic procedure is as follows:

1. For the chosen temporal resolution, select a suitable censor [mm] and apply it to the observed rainfall time-series by setting rainfall amounts below the censor to zero, and reducing rainfall amounts over the censor by the censor amount.

2. Fit the mechanistic rainfall model to the censored rainfall by aggregating the censored time-series to a range of temporal scales and calculating summary statistics as necessary for model fitting.

3. Simulate synthetic rainfall time-series at the same resolution as the training data in Step 1 and sample annual maxima.

4. Restore the censor to the simulated annual maxima and plot against the observed.

## 4. Model structure and selection

Mechanistic point process rainfall models, first developed by Rodriguez-Iturbe, Cox & Isham (1987) exist in various forms, although all models are formulated around two key assumptions about the rainfall generating process. Firstly, rainfall is





assumed to arrive in rain cells following a clustering mechanism within storms. Secondly, the total rainfall within cells is represented by a pre-specified rainfall pattern which describes the rain cell duration and amount. The continuous time rainfall is the summation of all rainfall amounts in time Δt. Most models assume rectangular pulses to describe rainfall amount and duration, although alternative patterns have included a Gaussian distribution (Northrop and Stone 2005) and instantaneous pulses (Cowpertwait et al. 2007, Cowpertwait et al. 2011, Kaczmarska et al. 2014). In this latter formulation, pulses are

assumed to arrive according to a Poisson process within cells, with each pulse representing an amount with zero duration. The

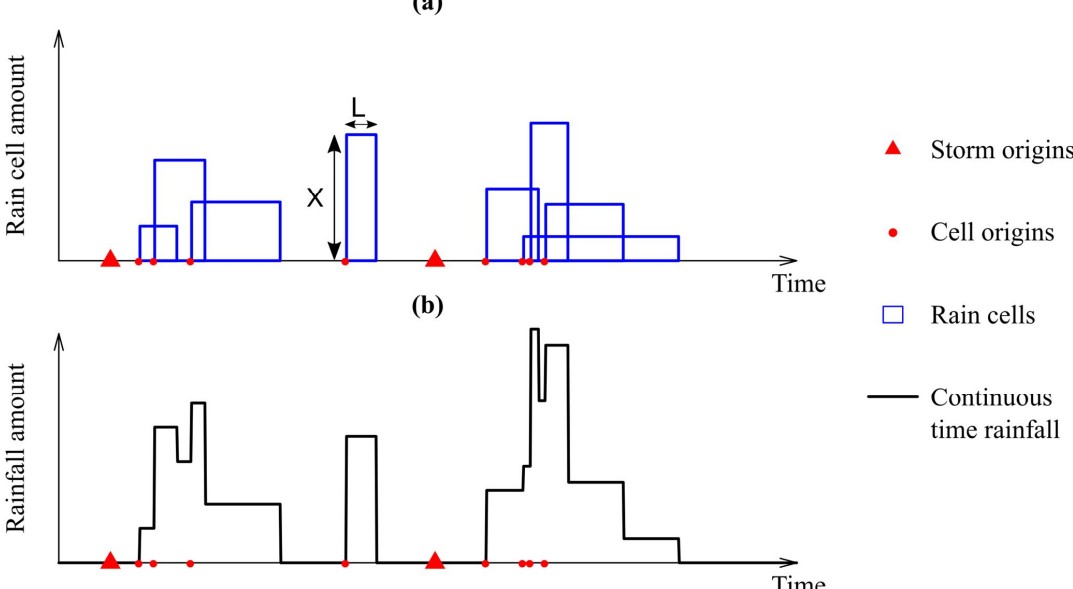

**Figure 2** Rainfall generation mechanism for mechanistic stochastic models with rectangular pulses. Panel (a) shows the arrival of storms and cells. Raincell intensity defines the height of each cell (X), and duration the length (L). Panel (b) shows the unobserved continuous-time rainfall time-series derived from the superposition of cells shown in (a).

continuous-time rainfall is therefore the summation of all pulse amounts in time Δt.

In the original form of the model, storms arrive according to a Poisson process with rate λ, and terminate after an exponentially distributed period with rate γ. The arrival of rain cells within storms follows a clustering mechanism which defines a secondary

Poisson process with rate β. Two clustering mechanisms are specified by Rodriguez-Iturbe, Cox & Isham (1987): the first is the Neyman-Scott mechanism in which the time intervals between storm and cell origins are assumed to be independent and identically distributed random variables; the second is the Bartlett-Lewis mechanism in which the time intervals between successive cell origins are independent and identically distributed random variables. In each case, the time intervals are assumed to be exponentially distributed. Rain cell profiles are rectangular with heights X for amounts, and lengths L for

durations. Both X and L are assumed to be independent of each other and follow exponential distributions with parameters η and $1/\mu_x$ respectively. See Fig. 2 for a graphical illustration of the continuous-time rainfall generation process.



The original Neyman-Scott and Bartlett-Lewis rectangular pulse models (NSRP and BLRP respectively) with exponential cell depth distributions are the most parsimonious models, each having only 5 parameters (see Table 1). A limitation of these models is that their simplicity implies all rainfall – stratiform, convective, and orographic - has the same statistical properties.

On the assumption that rainfall may derive from different storm types, in particular convective and stratiform, it is physically

**Table 1 Model parameters for original and randomized BLRP models and the original NSRP model.**

| | Units | BLRP | NSRP | BLRPR$_\eta$ | BLRPR$_X$ |
|---|---|---|---|---|---|
| Storm arrival rate | hr$^{-1}$ | $\lambda$ | $\lambda$ | $\lambda$ | $\lambda$ |
| Cell arrival rate | hr$^{-1}$ | $\beta$ | $\beta$ | $\{\beta\}^{(1)}$ | $\{\beta\}$ |
| Ratio of cell arrival rate to cell duration | - | - | - | $\kappa = \beta / \eta$ | $\kappa = \beta / \eta$ |
| Mean cell depth | mm hr$^{-1}$ | $\mu_x$ | $\mu_x$ | $\mu_x$ | $\{\mu_x\}$ |
| Ratio of mean cell depth to cell duration | mm | - | - | - | $\iota = \mu_x / \eta$ |
| Ratio of standard deviation to the mean cell depth | - | $r = \sigma_x / \mu_x$ | $r = \sigma_x / \mu_x$ | $r = \sigma_x / \mu_x$ | $r = \sigma_x / \mu_x$ |
| • Expected square of the cell depth $^{(2)}$ | mm$^2$ hr$^{-2}$ | $\{\mu_{x^2}\}$ | $\{\mu_{x^2}\}$ | $\{\mu_{x^2}\}$ | $\{\mu_{x^2}\}$ |
| • Expected cube of the cell depth for inclusion of skewness in the objective function $^{(2)}$ | mm$^3$ hr$^{-3}$ | $\{\mu_{x^3}\}$ | $\{\mu_{x^3}\}$ | $\{\mu_{x^3}\}$ | $\{\mu_{x^3}\}$ |
| Cell duration parameter | hr$^{-1}$ | $\eta$ | $\eta$ | $\{\eta\}$ | $\{\eta\}$ |
| • Gamma scale parameter for $\eta$ | - | - | - | $\nu$ | $\nu$ |
| • Gamma shape parameter for $\eta$ | hr | | | $\alpha$ | $\alpha$ |
| Storm duration parameter | hr$^{-1}$ | $\gamma$ | - | $\{\gamma\}$ | $\{\gamma\}$ |
| Ratio of storm duration to cell duration | - | | | $\varphi = \gamma / \eta$ | $\varphi = \gamma / \eta$ |
| Mean number of cells per storm | - | - | $\mu_c$ | - | - |
| Number of parameters: exponential cell intensity | - | 5 | 5 | 6 | 6 |
| Number of parameters: gamma cell intensity | - | 6 | 6 | 7 | 7 |

NOTES:

1. Parameters in curly brackets {} are not included in the objective function (see Sect. 5.2).

2. For the two parameter gamma cell depth distribution, the expected square and cube of the cell depth ($\mu_{x^2}$ and $\mu_{x^3}$) are calculated from the standard deviation ($\sigma_x$) and mean ($\mu_x$) of the cell depth. In practice it is the ratio of these (r) which is parameterized enabling calculation of $\mu_{x^2}$ and $\mu_{x^3}$. For both the exponential and gamma distributions, $\mu_{x^2} = f_1 \mu_x^2$ and $\mu_{x^3} = f_2 \mu_x^3$ where $f_1 = 1 + r^2$ and $f_2 = 1 + 3r^2 + 2r^4$. Because the exponential distribution is a special case of the gamma distribution where r is equal to 1, $\mu_{x^2} = 2\mu_x^2$ and $\mu_{x^3} = 6\mu_x^3$. Therefore it is not necessary to parameterize r for the exponential distribution, meaning the exponential versions of these models require 1 parameter less with r set to 1 in calibration.

more appealing to allow the statistical composition of rainfall models to vary between storms.

Two different approaches have been developed to accommodate the simulation of different rainfall types with rectangular pulses. For the Neyman-Scott model, concurrent and superposed process have been developed in generalised (Cowpertwait 1994) and mixed (Cowpertwait 2004) rectangular pulse models respectively. Both models enable explicit simulation of

multiple storm types, although their increased parameterization and consequent impact on parameter identifiability means that it is undesirable to simulate more than two storm types. For the Bartlett-Lewis model, randomization of the rain cell duration parameter η (Rodriguez-Iturbe et al. 1988, Onof and Wheater 1993, 1994b) with a Gamma distribution allows all storms to be drawn from different distributions. Because rain cell durations are assumed to be exponentially distributed, rain cells with high



values of η are more likely to be shorter in duration, and those with low values of η will typically have longer durations.

Additionally the rate at which rain cells arrive, and the storm durations, are defined in proportion to η by keeping the ratios β/η and γ/η constant (equal to κ and φ respectively). This means that typically, shorter storms will comprise shorter rain cells with shorter rates of arrival and the opposite for longer storms, which is characteristic of the differences between convective and stratiform rainfall.

The modified (random η) Bartlett-Lewis model (see BLRPR$_\eta$ in Table 1) of Onof & Wheater (1993, 1994b) is the most parsimonious of the model structures able to accommodate multiple storm types comprising a minimum of 6 parameters for the exponential version. The modified (random η) Neyman-Scott model has the same number of parameters as the modified Bartlett-Lewis model, but because there is no evidence that it has any advantage over the latter it is excluded from this study. The updated random η Bartlett-Lewis model with mean cell depth μ$_x$ also randomised (see BLRPR$_X$ in Table 1) requires fewer

parameters than its instantaneous pulse counterpart and the same number of parameters as the modified BLRPR$_\eta$ model. Structurally, it is identical to the modified model, although μ$_x$ is also allowed to vary randomly between storms by keeping the ratio ι = μ$_x$/η constant.

Because the Neyman Scott and Bartlett Lewis clustering mechanisms are considered to perform equally well, model selection is limited to the most parsimonious model structures within the Bartlett-Lewis family of models: the original model (BLRP),

the linear random parameter model (BLRPRη) and the linear random parameter model with randomized μ$_x$ (BLRPR$_X$). Hereafter, these models are referred to as BL0, BL1 and BL1M respectively. For the models used in this study, it is assumed that rain cells start at the storm origin.

## 5. Data and model fitting

### 5.1 Data selection

Estimation of fine–scale extremes with censored rainfall simulation is performed on two gauges: Atherstone in the UK and Bochum in Germany. Atherstone is a tipping bucket raingauge (TBR) operated and maintained by the Environment Agency of England. The record duration is 48 years from 1967–2015 , with one notable period of missing data from January 1974– March 1975. This site was selected from all TBRs for the Environment Agency's Midlands Region on the basis that the number of Environment Agency quality flags highlighted as "good" in the record is greater than 90%, and the number of "suspect"

flags less than 10% (92.3% and 6.7% respectively). Between the 8[th] February 1981 and after 20[th] November 2003 the gauge resolution is 0.5 mm. Before and after this period it is 0.2 mm. In the period before the 8[th] February 1981, the TBR record includes a number of observations of 0.1 mm at precisely 09:00:00. It is assumed that these are manual observations to correct the rain gauge totals to match with check gauge totals following quality checks of the data.

Bochum is a Hellmann raingauge operated and maintained by the German Meteorological Service. It uses a floating pen

mechanism to record rainfall on a drum or band recorder with a minimum gauge resolution of 0.01 mm. The duration is 69



years from 1931–1999, and the data are aggregated to a minimum temporal resolution of 5 minutes. These sites are selected to represent rainfall in different geographical regions obtained using different measurement techniques. Figure 3 shows the locations of these two gauges.

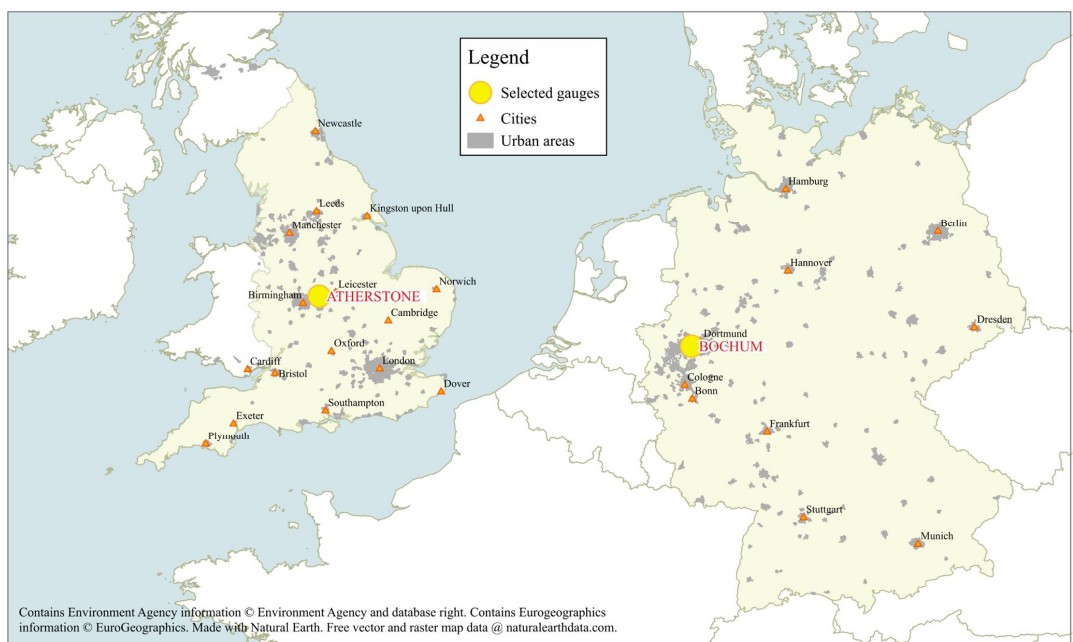

**Figure 3 Plan showing the location of the UK and German rain gauges used in this study.**

### 5.2 Parameter estimation

Model fitting is performed in the R programming environment (R Core Team 2016) using an updated version of the MOMFIT software developed by Chandler, Lourmas & Jesus (2010) for the UK Government Department for the Environment, Food and Rural Affairs (DEFRA) research and development project FD2105 (Wheater et al. 2007a, 2007b). In this software, parameter estimation is performed using the generalised method of moments (GMM) with weighted least squares objective function:

$S(\theta|T) = \sum_{i=1}^{k} \omega_i [t_i - \tau_i(\theta)]^2$. The reader is referred to Wheater et al. (2007b, Appendix A) for a detailed explanation of the fitting methodology.

The GMM is preferred for mechanistic rainfall models because the complex dependency structure and marginal distribution of aggregated time-series makes it very difficult to obtain a useful likelihood function (Rodriguez-Iturbe et al. 1988). In this procedure, the difference between observed and expected summary statistics of the rainfall time-series at a range of temporal scales is minimised giving an optimal parameter set $\theta$ where: $t = (t_1 \ldots t_k)'$ is a vector of k observed summary statistics, $\boldsymbol{\tau(\theta)} = (\tau_1(\theta) \ldots \tau_k(\theta))'$ is a vector of k expected summary statistics which are functions of $\boldsymbol{\theta} = (\theta_1 \ldots \theta_p)'$, i.e. of the vector of p model parameters for which analytical expression are available. The i-th summary statistic is weighted according to the inverse of its observed variance $\omega_i = 1/var(t_i)$ which has been shown to provide optimal weights for the GMM (Chandler et al. 2010). Other

weights can be applied allowing the user to influence the dominance of specific rainfall properties, although for unbiased

estimates of the summary statistics the weights must be independent of the model parameters and the data (Wheater et al.

2007b).

Typically, the vector of observed summary statistics T comprises the mean, variance, auto-correlation and proportion of dry

periods for temporal scales between 1 and 24 hours. Prior to model fitting and to allow for seasonality, summary statistics are

calculated for each month over the record length and pooled between months. For each month, the pooled statistics are used

to estimate the covariance matrix of model parameters required for parameter uncertainty estimation, and the mean of the

monthly statistics. Therefore 12 parameter sets are obtained for the whole year.

Model parameters are estimated using two minimisation routines. First, Nelder-Mead optimisations are performed on random

perturbations around user-supplied parameter values to identify promising regions of the parameter space. Following a series

of heuristics to identify the best performing parameter set, random perturbations around these values are used as new starting

points for subsequent Newton type optimisations. The parameter set with the lowest objective function is the best performing

and selected for that month. Following the approach employed by Kaczmarska (2013) to obtain smoothly changing parameters

throughout the year, this two-step optimisation is only applied to one month. Subsequent parameter estimation is based on a

single Newton type optimisation using the previous month's estimate as the starting point. Testing of this approach has shown

that when the parameters are well identified the same seasonal variation is achieved regardless of the starting month. GMM

parameter estimates are multivariate normally (MVN) distributed where the optimal parameter set is the mean, and the

covariance matrix is estimated from the Hessian computed by the optimisation routine (Wheater et al. 2007b). The MVN

distribution of model parameters is used to estimate 95% confidence intervals on the parameter estimates. If parameter

uncertainty is not estimated in model fitting it is indicative that the parameters are poorly identified.

### 5.3  Experimental design

Initial experiments with the coefficient of skewness and proportion of dry periods included in model fitting for censored data

were limited by the inability to obtain well identified parameters for some or all months. While good model fits were obtained

for some low censors, extreme value estimation continued to be understated. On the basis that censoring is a new approach to

enhance the estimation of rainfall extremes, skewness is not considered to be an important fitting statistic for censored

simulations. Furthermore, because censored models cannot be used to generate continuous time-series of the sort which may

be used for hydrological modelling, the proportion of dry periods is also considered to be unimportant for censoring.

Consequently for censored model calibration, the choice of fitting statistics is reduced to the 1 hour mean, the coefficient of

variation and lag-1 autocorrelation of the rainfall depths at the censor resolution, and the 6 and 24 hour resolutions. Again, to

ensure well identified model parameters for the Atherstone dataset, it was necessary to extent the choice of fitting statistics to

include the 1 hour statistics for 5 minute simulations. This was neither necessary for 15 and 60 minute simulations at

Atherstone, nor the Bochum dataset.



For all simulations the fitting window is widened to 3 months, hence for any given month the models are fitted to data for that month, together with the preceding and following months. This approach is used to increase the data available for fitting the models when censoring on the basis that censoring removes data which would otherwise be used in fitting. Tests have shown that widening the fitting window from one to three months has the effect of smoothing the seasonal variation in model

parameters and improving parameter identifiability. There is also negligible impact on the estimation of summary statistics and extremes under the model parameters.

For the two randomized models, BL1 and BL1M, the Gamma shape parameter α is constrained to a fixed value in calibration and simulation. The Gamma shape parameter α is an insensitive model parameter and can take any value within a very large range without significant impact on the estimation of summary statistics or extremes (see Appendix A). For the BL1 model,

parameterization without an upper bound on α often results in poor identifiability with parameter estimates in the thousands to tens of thousands. For the BL1M model, α is typically better identified than for BL1 with a tendency to move towards the lower boundary. In order to avoid having infinite skewness, α must be greater than 4 for the BL1 model and 1 for the BL1M model (see Kaczmarska, Isham & Onof (2014) and references therein for discussion on these criteria). Therefore, by fixing α at 100 for the BL1 model and 5 for the BL1M model, the number of parameters to be identified for these models is reduced

by one. All models are fitted using the exponential distribution for mean cell depth. This further reduces the number of model parameters to be fitted for both uncensored and censored models, therefore in all cases the ratio of standard deviation to the mean cell depth ($r = \sigma_x/\mu_x$) is fixed at 1. Fitted model parameters are presented in Appendix B for 5 and 15 minute rainfall at both sites for uncensored and censored rainfall using censors selected in Sect. 6.2 (Table 2).

## 6.   Results

### 6.1   Extreme value estimation

Rainfall extremes are estimated from the models by sampling annual maxima directly from simulations. For each model fitted to uncensored data, 100 realisations of 100 years duration are simulated using parameters randomly sampled from the multivariate normal (MVN) distribution of model parameters. This allows model parameter uncertainty to be represented in the spread of the MVN extreme value estimates (hereafter referred to as MVN realisations), covering the full range of

observations. For extrapolation, rainfall extremes are also estimated from one realisation of 10,000 years duration simulated using the mean of the MVN parameter distribution (hereafter referred to as the optimal estimates). By extending this simulation to 10,000 years duration, extreme value estimation up to approximately the 0.001 AEP (1000 year return level) may reasonably be expected to be stable.

Extreme value estimation for the censored calibrations is shown in Figs. 4, 5 and 6 for 5, 15 and 60 minutes temporal resolutions

respectively. The top three panels (a–c) in each figure show the results for Bochum, and the bottom three panels (d–f) the results for Atherstone, with observed and simulated annual maxima plotted using the Gringorten plotting positions.





**Figure 4 Extreme value estimation at 5 minute resolution. Optimal realisations (opt. AM) are shown with solid lines and the mean of the MVN realisation (mvn. AM) are shown with dashed lines.**



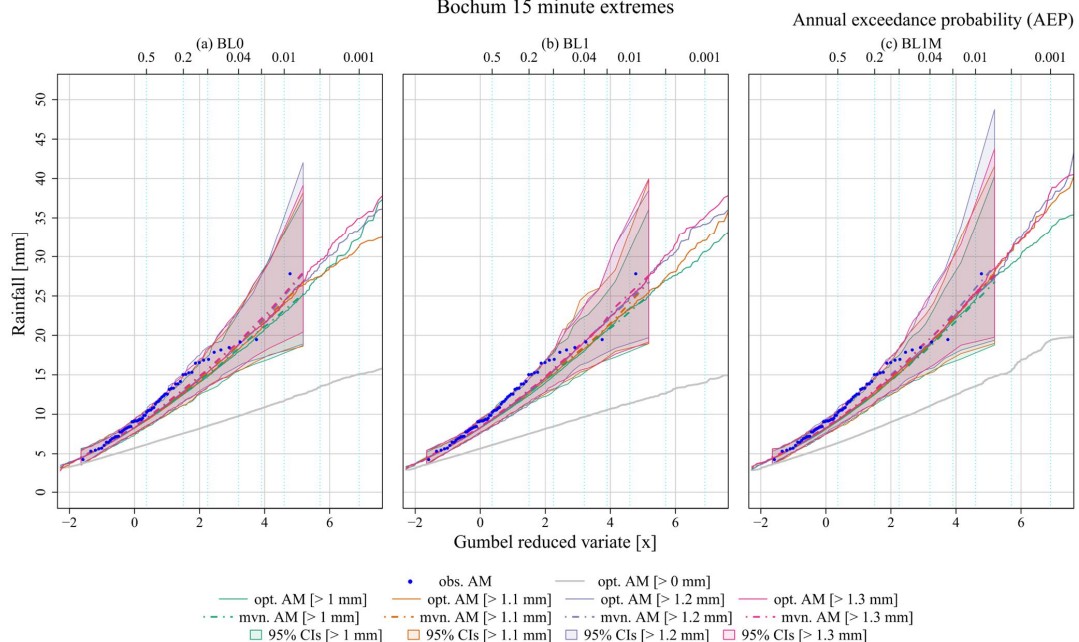

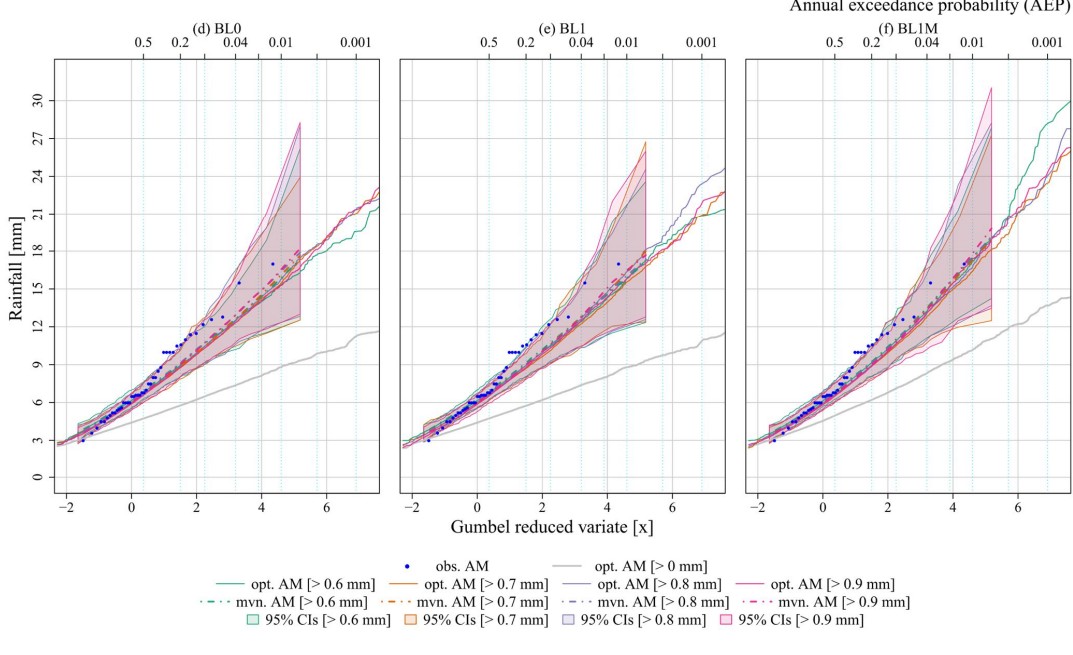

**Figure 5 Extreme value estimation at 15 minute resolution. Optimal realisations (opt. AM) are shown with solid lines and the mean of the MVN realisation (mvn. AM) are shown with dashed lines.**





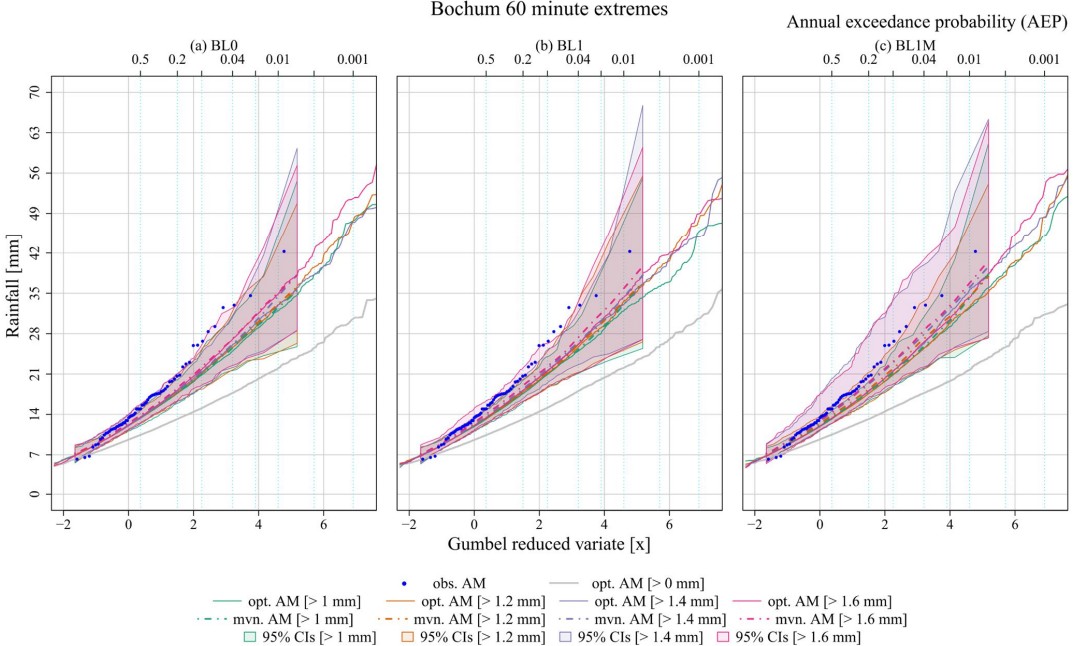

**Figure 6 Extreme value estimation at 60 minute resolution. Optimal realisations (opt. AM) are shown with solid lines and the mean of the MVN realisation (mvn. AM) are shown with dashed lines.**

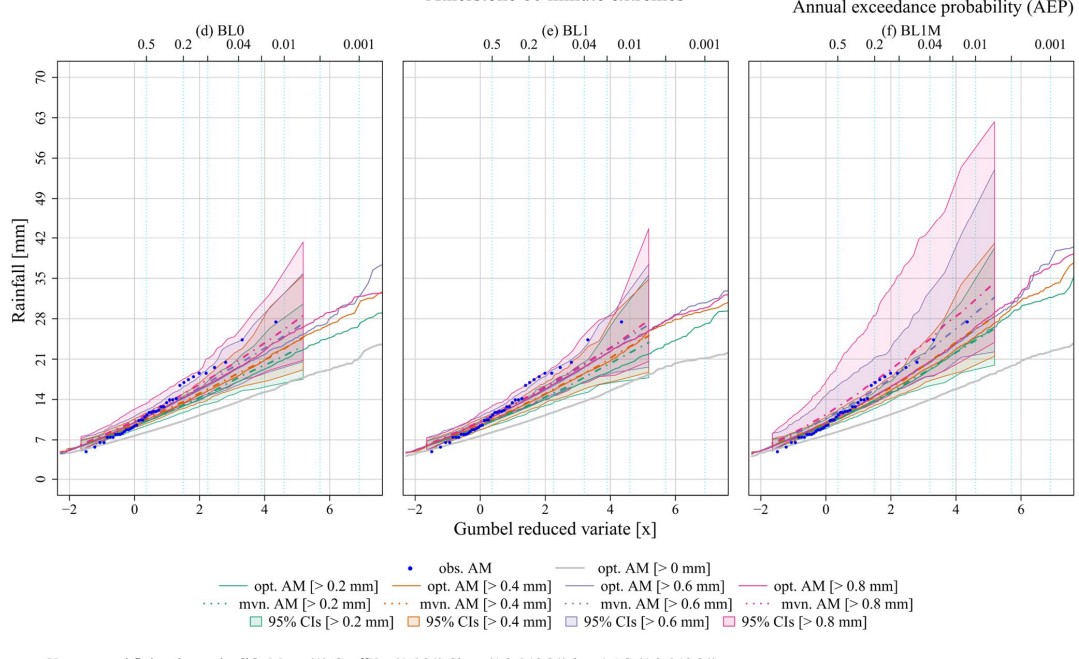



All plots show the equivalent extreme value estimates obtained without censoring obtained by simulating one realisation of 10,000 years duration with the optimal parameter set. Upper limits on censoring were identified when model parameterization noticeably deteriorated resulting in the mean of the MVN realisations to deviate away from the optimal. Results presented are limited to the 4 highest censors with behavioural parameterization, together with 95% confidence intervals derived from MVN realisations.

All censored models have significantly improved the estimation of extremes at each site and scale with very good estimation by all three model variants particularly at the 5 and 15 minute scales. At these scales, the estimation of extremes with the 4 censors presented has approximately converged on the observations. At the 60 minute scale there is notable improvement in the estimation of extremes with some convergence in estimation with increasing censors, although there is continued underestimation of the observed. The 95% confidence intervals by all censored models broadly bracket the observations and

are largely unvaried with increasing censors, other than with the BL1M model at the 60 minute resolution.

At the 5 minute scale, estimation has converged on the observations with censors between 0.5 and 0.65 mm at Bochum, and between 0.6 and 0.75 mm at Atherstone. For all three models there is slight underestimation of extremes higher than approximately the 0.1 annual exceedance probability (AEP), although the BL1M model accurately estimates the highest observed extreme at both sites. At the 15 minute scale, convergence at Bochum has occurred for censors between 1.0 mm and

1.3 mm, while at Atherstone convergence has occurred for censors between 0.6 mm and 0.9 mm. As for the 5 minute resolution models, the BLIM model appears to perform slightly better than the BL0 and BL1 models, resulting in improved estimation of the highest observed extremes and elevated estimates of the 0.001 AEP at both sites. At the 60 minute resolution, there is good convergence in estimation for all three models at Bochum, and the BL1M model at Atherstone. However, extreme value estimation with the BL0 and BL1 models at Atherstone is more widely spread across the applied censors. For the BL0 and

BL1 models, the 0.2 mm censor results in much lower estimates than the three higher censors, although the mean of the MVN realisations for the 0.6 and 0.8 mm censors are starting to deviate away from the optimum realisation. For the BL1M model, there is good convergence between the optimal realisations with each censor, although the mean of the MVN estimates for the 0.6 and 0.8 mm censors have significantly deviated from the optimum.

The mean of the MVN realisations for the BL1M model at Atherstone with the 0.6 and 0.8 mm censors diverges from the

optimum because of the generation of unrealistic extremes. This divergence is also observable in the larger spread of 95% confidence intervals over 100 realisations. While it has been possible to fit the model, Fig. 7 shows that as censoring has increased to 0.8 mm, confidence intervals on model parameters have widened for several months of the year, notably January, February and June. When sampling from the MVN distribution in simulation, these large confidence intervals mean that there is a high chance of sampling parameters which deviate significantly from the mean of the distribution thereby giving rise to a

wide spread in extreme value estimates. These large confidence intervals indicate that the parameters are poorly identified and therefore the model error is too large for reliable simulation.





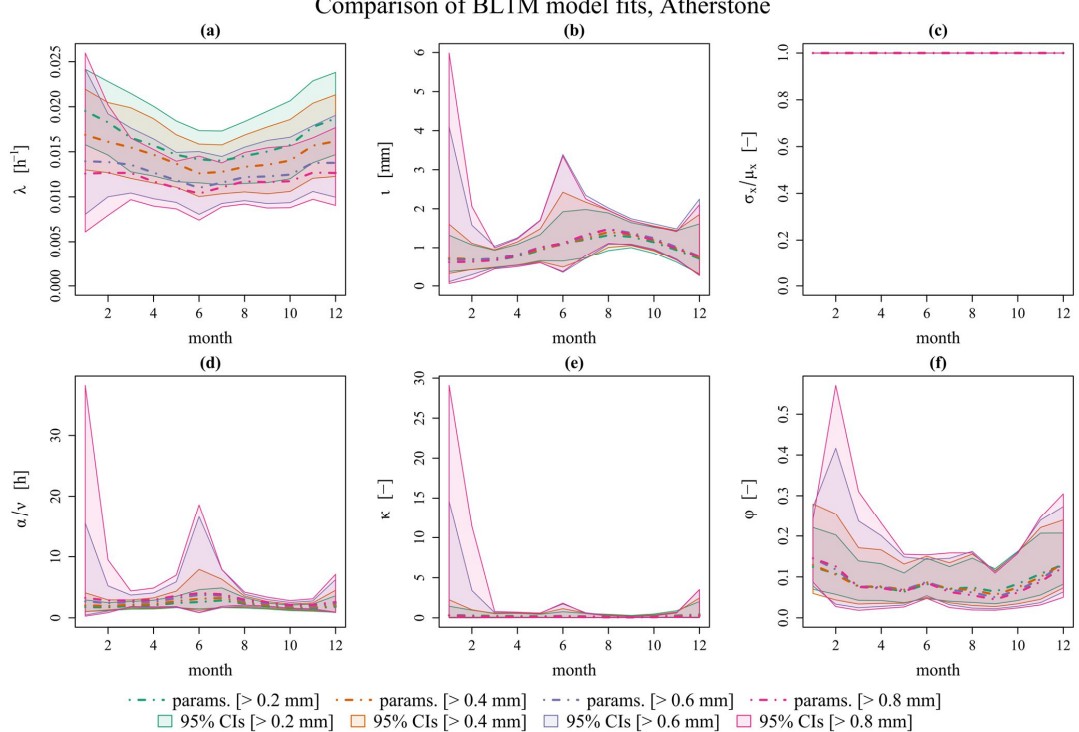

**Figure 7 Comparison of censored BL1M model parameters for Atherstone 60 minute data. Optimal parameter estimates (params.) are shown with dot-dashed lines, and parameter uncertainty is represented with 95% confidence intervals.**

### 6.2 Validation

The rainfall extremes presented in Sect. 6.1 have been generated mechanistically using model parameters derived from central

moments of the censored rainfall time-series. While censored models cannot be used to simulate the whole rainfall hyetograph, it is important to ensure that the process by which the extremes are estimated is reliable. Therefore, model performance is validated in the usual way for this class of model by comparing the analytical summary statistics under the model parameters with the observations – here the observations are censored. The lowest censors presented in Figs. 4, 5 and 6 are selected for validation. No distinction is made between models in this choice, although it is recognised that there is some variation in the

extreme value performance of specific censors between model types. Given the spread of estimates at Atherstone for the 60 minute resolution, validation is based on the 0.6 mm censor instead of the lowest which would be 0.2 mm. See Table 2 for censor selection at each site and scale.

**Table 2 Censor selection for model validation.**

|  | 5 minutes | 15 minutes | 60 minutes |
|---|---|---|---|
| Bochum | 0.5 mm | 1.0 mm | 1.0 mm |
| Atherstone | 0.6 mm | 0.6 mm | 0.6 mm |





**Figure 8 Seasonal variation in mean, coefficient of variation and lag-1 autocorrelation for selected optimal censors, observed vs. estimated.**





### 6.2.1    Replication of fitting statistics

Figure 8 shows the seasonal variation in mean, coefficient of variation and lag-1 autocorrelation for all three models with the selected censors in Table 2. The plots show the estimated summary statistics calculated using the optimum parameter estimates, together with 95% confidence intervals obtained by randomly sampling 100 parameter sets from the multivariate normal distribution for model parameters. Because models are fitted over 3 monthly moving windows, estimated summary statistics are compared with summary statistics for censored observations for the same periods. Fitting statistics for the 6 and 24 hour scales are not shown. The limits on the vertical Y axes are optimized at each site and scale, therefore the reader is advised to pay careful attention to the scales when comparing summary statistics.

All six models perform very well with respect to replicating the summary statistics used in fitting with the 95% confidence intervals comfortably bracketing the observations. The estimated summary statistics are very close to the observed with all models performing equally well. The plots in Fig. 8 demonstrate that the models are able to reproduce the censored fitting statistics, confirming reliability of the process.

### 6.2.2    Replication of statistics not used in fitting

Figure 9 shows the seasonal variation in the coefficient of skewness and proportion of wet periods for all three models with the selected censors in Table 2. Both of these statistics were excluded from model fitting for censored simulations, although are generally considered to be important fitting statistics. All the models over-estimate the skewness coefficient with observations for all months falling outside the 95% confidence intervals. This is not an unexpected result given that censoring effectively truncates the thin tail of the rainfall amounts distribution which will significantly change its shape. Because this truncation is not replicated in the analytical equations the models are unlikely to be able to replicate this behaviour using the observations provided.

The ability of the models to reproduce the proportion of wet periods is generally better than their ability to reproduce skewness. At the 5 minute resolution for Bochum, the 95% confidence intervals comfortably bracket the observations between the months of May and October, although there is over-estimation in the other months and for all months at the 15 and 60 minute scales. At Atherstone, there is good representation of the proportion of wet periods at the 15 and 60 minute scales, although over-estimation at the 5 minute scale.





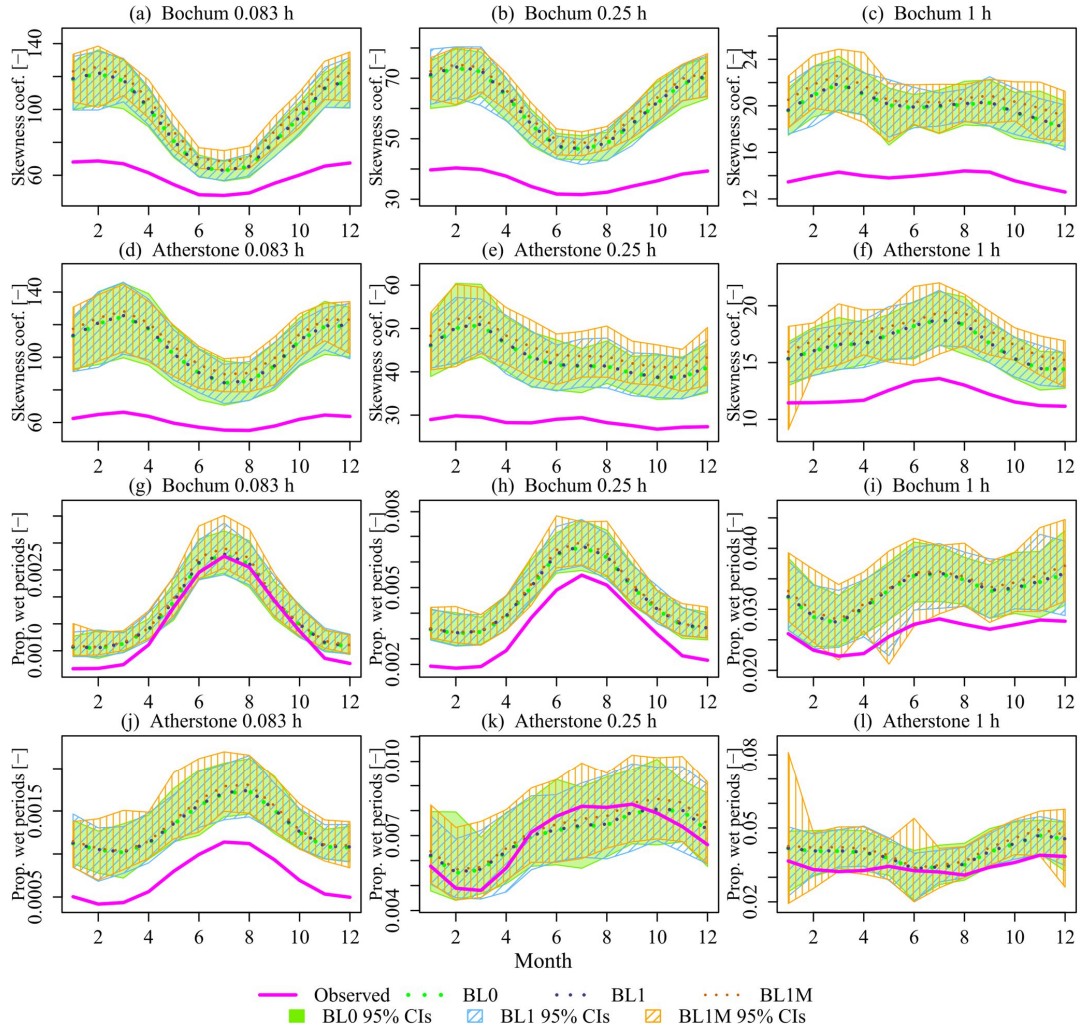

**Figure 9 Seasonal variation in skewness coefficient and proportion of wet periods for selected optimal censors, observed vs. estimated**

### 6.2.3    Parameter identifiability

Figure 10 shows profile objective function plots for the BL0 model fitted to Bochum 5 minute data with a 0.5 mm censor for the month of January. In the case of the specific month and model presented, the logarithm of parameters $\mu_x$, $\beta$ and $\gamma$ are well identified. Clear minima are also well identified for $\lambda$ and $\eta$, although the profile objective functions for these parameters indicate extended flat regions less than and greater than the minima respectively. All profiles show the parameters are well identified at the 95% confidence level which suggests that the optimum minima are identified. Using the two–stage  fitting methodology outlined in Sect. 5.2, identical parameterizations were obtained for different starting months in most cases. This is indicative that parameterization is robust and that the same minima are always identified.





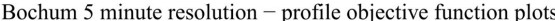

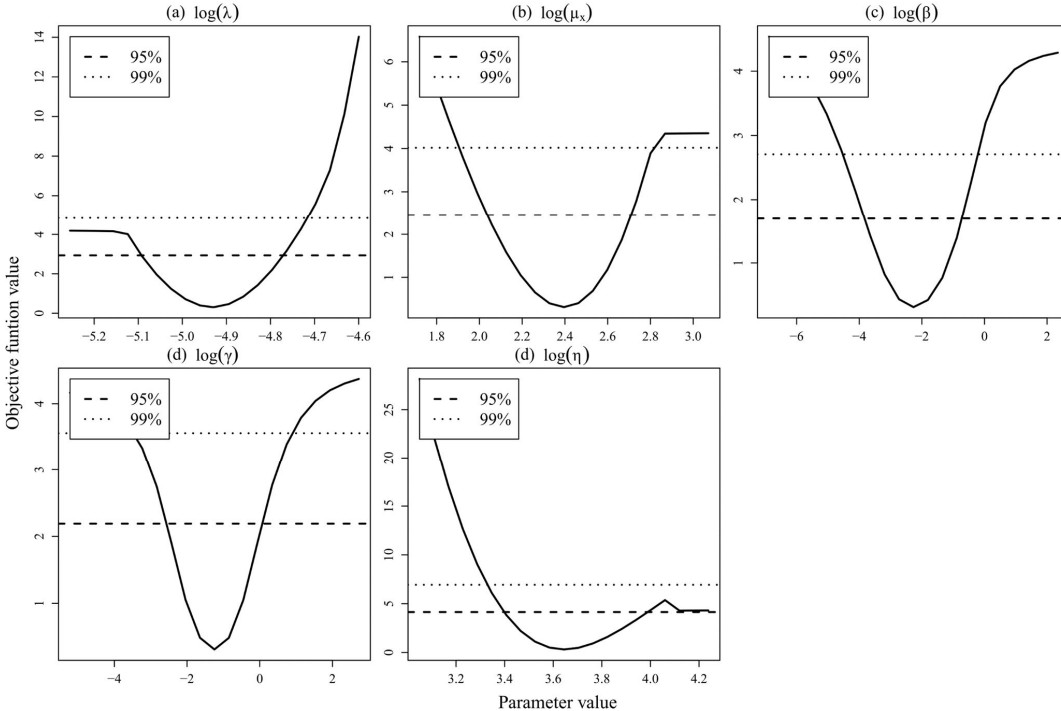

**Figure 10 Profile objective function plots for logarithm of the BL0 model parameters: model fitted to Bochum 5 minute data with 0.5 mm censor for January.**

## 7. Discussion on censor selection

The censors selected for validation in Table 2 were chosen based on their extreme value performance. For the estimation of extremes at other locations, it would be preferable to have a set of heuristics to guide censor selection. The following discussion of extreme value estimation performed in this study is intended to guide practitioners in the application of censored modelling.

### 7.1 Stability of confidence intervals

Upper limits on censoring were identified where model parameters were either poorly identified or the mean of the MVN realisations deviated significantly from the observations. The onset of this effect was observed in Fig. 6 for estimation of hourly extremes at Atherstone with the BL1M model. Figure 11 shows the change in 95% confidence intervals and the mean of the MVN realisations obtained with behavioural and non-behavioural censored models for 15 minute data at Bochum and Atherstone. The comparison is made between extremes for the selected censors given in Table 2 (1.0 mm and 0.6 mm respectively) and extremes from higher censors (1.5 mm and 1.0 mm respectively).



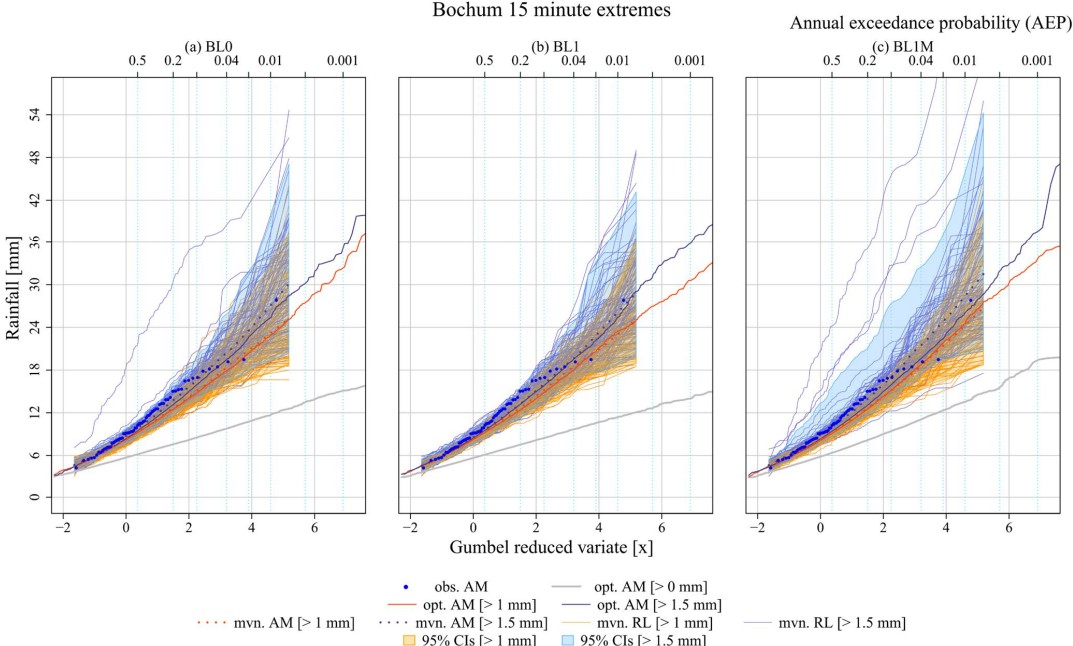

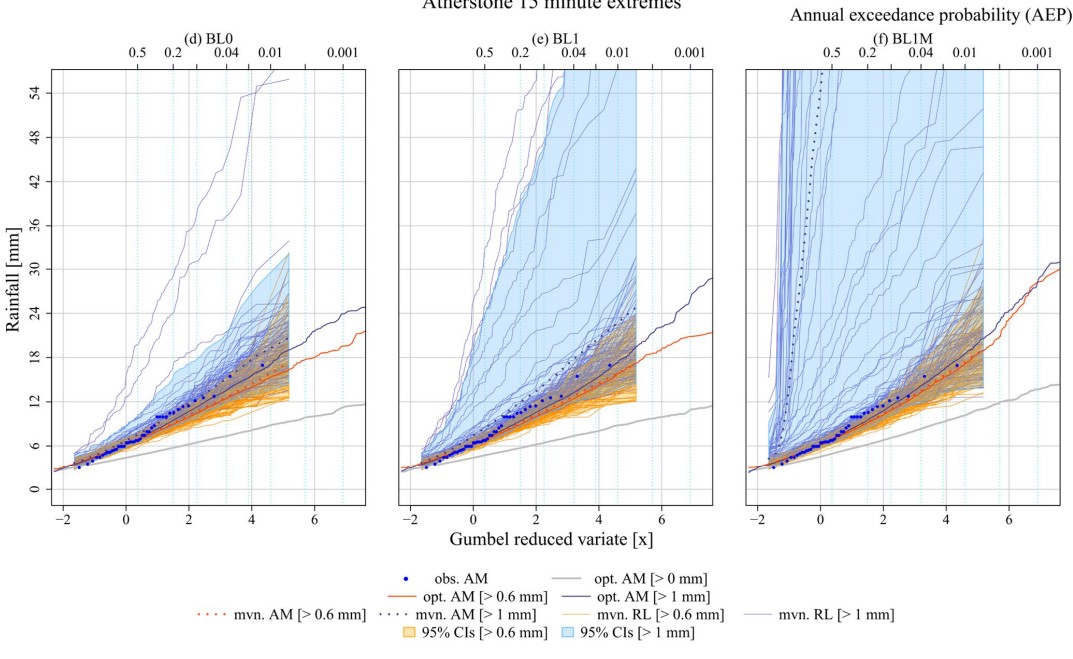

**Figure 11 Change in 95% confidence intervals and mean of the MVN realisations (MVN. AM) for Bochum and Atherstone 15 minute data with behavioural (> 1.0 mm and > 0.6 mm) and non-behavioural (> 1.5 mm and > 1.0 mm) censors. The distribution of individual MVN realisations (mvn. RL) are shown with grey lines.**

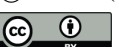



Confidence intervals on extreme value estimates for Bochum 15 minute rainfall obtained with censors from 1.0–1.3 mm, and for Atherstone with censors from 0.6–0.9 mm (Fig. 5), are broadly stable and unchanging. This is indicative that parameterization across each model variant and censor is good enabling robust estimation of extremes. As the censor at

Bochum is increased to 1.5 mm (Fig. 11, panels a–c), there is a noticeable increase in the upper confidence bound and the mean of the MVN realisations has diverged leading to over-estimation of the extremes. Increasing the censor at Atherstone to 1.0 mm has resulted in very significant widening of the confidence intervals and divergence of the mean of the MVN realisation (Fig. 11, panels d–f). In each case, this divergence results from the generation of unrealistic extreme value realisations which are shown in Fig. 11 with light grey lines.

While it has been possible to fit models to data with these high censors, examination of the parameter estimates and associated uncertainty reveals that parameter identifiability is reducing. Figure 12 shows the seasonal variation in estimates for the BL1M model parameters α/ν, κ and φ fitted to Bochum 15 minute data with a 1.5 mm censor. Parameters λ and ι are well identified with tight confidence brackets around the optimum, while r and α are fixed, therefore these parameters are not shown. Confidence intervals on α/ν, κ and φ are very large in the winter months indicating that identifiability of these parameters has

deteriorated. When sampling from the multivariate normal distribution for model parameters in simulation, these large uncertainties give rise to non-behavioural extreme value estimation. The same behaviour was observed for the BL1M model at Atherstone for 60 minute data as shown in Fig. 7.

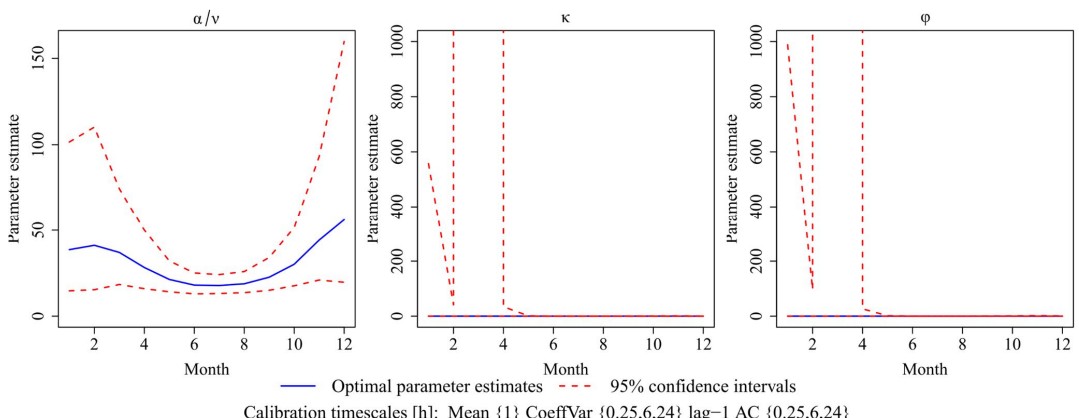

**Figure 12 Fitted model parameters for the BL1M model with 1.5 mm censor applied to Bochum 15 minute data.**

With the upper bound on censoring identified, the obvious question is how to identify a lower bound? The results presented in

Figs. 4, 5 and 6 suggest that there is convergence in the estimation of extremes with increasing censors. If so, when is the onset of convergence? Figure 13 shows the change in extreme value estimation with censor for 15 minute rainfall at Bochum (panels a–f) and Atherstone (panels g–l) for 0.1 and 0.04 AEPs.







**Figure 13 Variation in extreme value estimation with censor for 15 minute data at Bochum and Atherstone for two annual exceedance probabilities: 0.1 and 0.04. The spread of MVN realisations (mvn. RL) are shown in the box plots.**





At both locations, divergence in the mean of the MVN realisations and confidence intervals shown in Fig. 13 is easily identified
with the very large box-plot whiskers at 1.5 mm and 1.0 mm censors for Bochum and Atherstone respectively. The plots for
Bochum also show a large spread in the extreme realisations with a 1.4 mm censor for the BL1M model suggesting that
parameter identifiability is deteriorating at this censor.

At Atherstone, there is clear evidence of convergence in estimation between censors 0.5–0.9 mm. However, convergence is
less obvious at Bochum. At Bochum, there is continual improvement in extreme value estimation with the increasing censors,
although there is a perceptible reduction in improvement with each successive increase in censor. For censors of 0.7 mm and
above, all model realisations bracket the observed extremes (horizontal dashed blue line), which is also true for censors above
0.5 mm at Atherstone. Therefore, ranges may be identified at both sites for censors which may be considered to give
satisfactory estimation of extremes: 0.7–1.3 mm at Bochum and 0.5–0.9 mm at Atherstone.

### 7.2 How much rainfall to censor?

In Sect. 7.1 we identify plausible censor ranges on the basis parameter stability and convergence of extreme value estimation.
However, this doesn't address the question of how much rainfall to censor? Because extremes are generated mechanistically,
we want to simulate the storm event hyetograph therefore it is in our interest to keep the censor low in relation to the rainfall
depth profile. The most basic check is that the minimum observed extreme (here designated as the smallest annual maxima) is
greater than the censor being used. This is true for all the sites and scales investigated in this study, with the lowest observed
annual maxima of 1.6 mm occurring at the 5 minute scale in Atherstone. However, this significantly exceeds the maximum
censor applied to 5 minute data at Atherstone, 0.75 mm (see Fig. 4), therefore it's unlikely that a well parameterized model
would be achieved.

Figure 14 shows the empirical cumulative distribution function (ECDF) plots for the above zero rainfall records at Bochum
and Atherstone aggregated to 5 and 15 minute resolutions. All the censors used for the estimation of fine–scale extremes in
Figs. 4, 5 and 6 are shown, with the top three censors highlighted magenta. The censors selected for model validation (Table 2)
are highlighted blue, and their quantiles listed separately in Table 3. The ECDF plots are truncated at the 99th percentile to aid
comparison of the applied censors, therefore the maximum rainfall is highlighted in red text on the right of each plot. For the
15 minute plots, the lower limits on censors identified in Sect. 7.1 are shown and highlighted green.

**Table 3 Quantile of the above zero rainfall amounts, proportion of maximum rainfall and number of independent
peaks per year for the selected censors given in Table 2.**

|  | Scale [mins] | Bochum | Atherstone |
| --- | --- | --- | --- |
| Quantile of rainfall amounts | 5 | 98.3% | 96.5% |
|  | 15 | 96.8% | 81.7% |
| Proportion of maximum rainfall | 5 | 3.0% | 5.7% |
|  | 15 | 3.6% | 3.5% |
| Number of independent peaks / year | 5 | 53 | 27 |
|  | 15 | 46 | 65 |





It can be seen from Fig. 14 that a substantial proportion of the above zero rainfall record is masked from the models with

censoring. At the 5 minute scale, the selected censor of 0.5 and 0.6 mm removes in excess of 98% and 96% of the above zero

rainfall from Bochum and Atherstone respectively. At the 15 minute scale, the selected censors of 1.0 and 0.6 mm remove in

excess of 96% and 81% respectively. These quantiles are high and support the hypothesis that mechanistic models may be

poor at estimating fine–scale extremes because the training data are dominated by low observations.

**Figure 14 Empirical Cumulative Distribution Function plots for Bochum and Atherstone rainfall aggregated to 5 and 15 minute temporal resolutions. The plots are limited to the 99th percentile rainfall and show the rainfall quantiles corresponding to the optimum censors used in the estimation of extremes in Figs. 4, 5, and 6.**




A striking difference in the ECDF plots for the two locations is the smoothness of the curves. The stepped nature of the Atherstone plots is very pronounced and reflects the resolution of the gauge: 0.5 mm between 1982 and 2003, and 0.2 mm before and after these dates. The stepped nature of the plots at Atherstone highlights that the selected censor quantiles (blue) are just greater than the 0.5 mm quantiles. We also know from Fig. 13 that a censor of 0.5 mm for 15 minute rainfall at

Atherstone would give very similar extreme value estimation to the selected 0.6 mm censor (highlighted green on Fig. 14 panel d). This implies that to improve the estimation of fine–scale extremes at Atherstone, it has been necessary to remove all observations which correspond with the gauge resolution.

While the proportion of rainfall observations removed prior to model fitting is large - over 90% and 80% for 5 and 15 minute rainfall from Bochum and Atherstone respectively - comparison with the maximum rainfall amounts and an assessment of the

number of independent peaks over the censor demonstrate that the censors are low (see Table 5). The proportion of the maximum observed rainfall is less than 6% in all cases which is very low considering that the maximum recorded rainfall across both sites and scales is just 27.9 mm for Bochum 15 minute rainfall. For a Peaks over Threshold extreme value analysis, the threshold is typically set so that between 3 and 5 independent peaks per year remain in the partial duration series. Using a temporal separation of 48 hours to define independence, the number of peaks per year retained after censoring is between 27

and 65 (Table 3). The actual number of peaks retained for fitting the Bartlett-Lewis models is much greater than this because there is no requirement to meet the independence criteria with mechanistic modelling.

## 8.   Further discussion

The estimation of rainfall extremes presented in this study using censored rainfall simulation is highly promising and offers an alternative to frequency techniques. The estimation of extremes at sub-hourly scales has far exceeded expectation with all

three models giving a very high level of accuracy across a range of censors. However, censoring uses rainfall models in a way they were never previously intended. Rainfall models have invariably been used for simulation of long duration time-series across a range of scales for input into hydrological and hydrodynamic models. Censored rainfall synthesis cannot be used in this way because only the heavy portion of the hyetograph is simulated.

The success of this research is to broaden the scope of mechanistic rainfall modelling and ask new questions of it. Mechanistic

models and related weather generators are very powerful at simulating key summary statistics for a range of environmental variables. An area where these models have consistently underperformed is the estimation of fine–scale extremes. Efforts to improve extreme value estimation at fine temporal scale have focussed on structural developments. But those developments have always been undertaken in the context of rainfall time-series generation. Continued underestimation at fine temporal scales has given rise to the notion that rectangular pulse models are potentially *"unsuitable for fine–scale data"* (Kaczmarska

et al. 2014, p.1985).



For effective scenario planning with hydrological models, good reproduction of rainfall time-series is necessary, with accurate estimation of key summary statistics. However, for assessment of extremes and estimation of storm profiles, good replication of rainfall central moments is arguably less important. The ability of the censored models to adequately reproduce the central moments used in calibration was checked to ensure that the process by which the extremes are constructed is reliable. Because rainfall over the censor is by definition coincident with rainfall below the censor, the censored models can be used to estimate uncensored extremes by simply restoring the censor to the estimates.

Extreme rainfall estimation with censoring across all models, scales and sites is significantly improved on that without censoring as shown in Figs. 4, 5 and 6. Up to approximately the 0.04 AEP, estimation is broadly equivalent across all models. For rarer events, the BL1M model appears to perform better than the other two at the 5 and 15 minute scales at Bochum and Atherstone by accurately estimating the highest observations at those scale. This improvement over the BL0 and BL1 models is significant in the event that extreme rainfall estimation is required beyond the range of observations. This is demonstrated in all 4 cases (5 and 15 minute scales at Bochum and Atherstone) with the higher estimation of extremes at the 0.001 AEP by the BL1M model compared with the other two. Below approximately the 0.04 AEP the differences in extreme rainfall estimation are so small that it is not possible to single out any one model as having the best overall performance, although for lower probability events the results suggest a preference for the BL1M model. This result supports the findings reported by Kaczmarska, Isham & Onof (2014) that the dependence structure between rain cell amounts and duration in the BL1M model is beneficial in estimating fine–scale extremes.

## 9. Conclusions

Censored rainfall synthesis using mechanistic pulse based models appears to offer an alternative approach to estimation of rainfall extremes and to frequency estimation techniques. The results presented in this paper show that the method has worked for single site data from two very different locations which has been collected using different gauging techniques. Consistency in the optimal censor identified for each location, and the stability of estimation across a range of censors gives confidence in the approach and supports the original hypothesis.

It is an obvious limitation of censoring that it cannot be used to obtain time-series of synthetic rainfall as is the principal application of mechanistic rainfall models. However, the intention of this research was to investigate if mechanistic models could be used for estimation of fine–scale extremes as an alternative to frequency techniques. The accuracy of estimates for sub-hourly rainfall extremes using all three model variants is very good, although the BL1M model appears to outperform the other two models at both sites for the 5 and 15 minute scales by accurately predicting the highest observed extreme.

Reducing parameterization by fixing the Gamma shape parameter $\alpha$ in the randomised models, and increasing the data for parameterization by widening the fitting window to 3 months has enabled the models to be fitted successfully to censored observations. It is likely that these aides to parameterization are necessary because censoring truncates the statistical

distribution of the training data. The analytical solutions in the models do not make this assumption, therefore a mismatch between the training data and the models arises with censoring. At low censors, truncation is minor and the analytical solutions in the models are able to make reasonable estimates of the fitting statistics. However, as the censor increases and the mismatch

grows a point is reached at which the analytical solutions are no longer able to estimate the fitting statistics causing deterioration in parameter identifiability.

A principal goal of this research was to improve the physical basis of short duration extreme rainfall estimation. This has been achieved by simulating storm profiles mechanistically in a way which mimics the phenomenology of rainfall generation. This has given rise to extreme rainfall estimation which may be described as a function of a set of model parameters with physical

meaning, e.g., the extreme rainfall quantile $z = F\{\lambda, \mu_x, \delta_x, \delta_c, \mu_c, \delta_s\}$ for the original Bartlett-Lewis model (See Appendix A for definitions of mechanistic model parameters). Future research is required to link model parameters to environmental covariates and spatial information. The latter may follow the regionalisation methodology of Kim et al. (2013).

Further research is also required to investigate the potential for incorporating censored modelling into a multi-model approach for synthetic rainfall generation. This may take the form of simulating the rainfall below the censor using a bootstrapping

approach similar to that in Costa, Fernandes & Naghettini (2015), or continuous simulation of uncensored rainfall with replacement of storms simulated using the censoring approach.

**Data availability**

The Atherstone tipping bucket raingauge dataset was obtained directly from the Environment Agency for England, UK. The data are not publicly accessible because they are used by the Environment Agency for operational purposes, but can be obtained

for non-commercial purposes on request. The Bochum dataset was obtained directly from Deutsche Montan Technologie and was recorded by the Emschergenossenschaft / Lippeverband in Germany. The data are not publicly accessible because they belong to the Emschergenossenschaft and Lippeverband public German water boards and are used for operational purposes.

**Appendix A: Bartlett-Lewis model parameter sensitivity and impact on extreme value estimation.**

To demonstrate the insensitivity of α for the randomised Bartlett-Lewis models, the BL1 and BL1M models were fitted to

Bochum 15 minute rainfall with changing constraints on α. The models were fitted using the 1 hour mean and the 0.25, 6 and 24 hour coefficient of variation, skewness coefficient and lag-1 autocorrelation. For the BL1 model, α is constrained between 4.1 (lower bound) and 5, 10, 25, 50, 75 and 100 (upper bounds). For the BL1M model, α is constrained between 5, 10, 25, 50, 75 and 100 (lower bounds) and infinity (upper bound). For the BL1 and BL1M models, α converges on the upper and lower bounds respectively, although because α is not held fixed parameter uncertainty is estimated. Parameter ranges are presented

in the parallel coordinate plots in Fig. A1 by sampling 1000 parameter sets from the multivariate normal distribution of model





parameters for 4.1 > α < 1 million. The parameter sets corresponding to α = 100 and α = 5 are shown for the BL1 and BL1M

models respectively with dashed magenta lines.

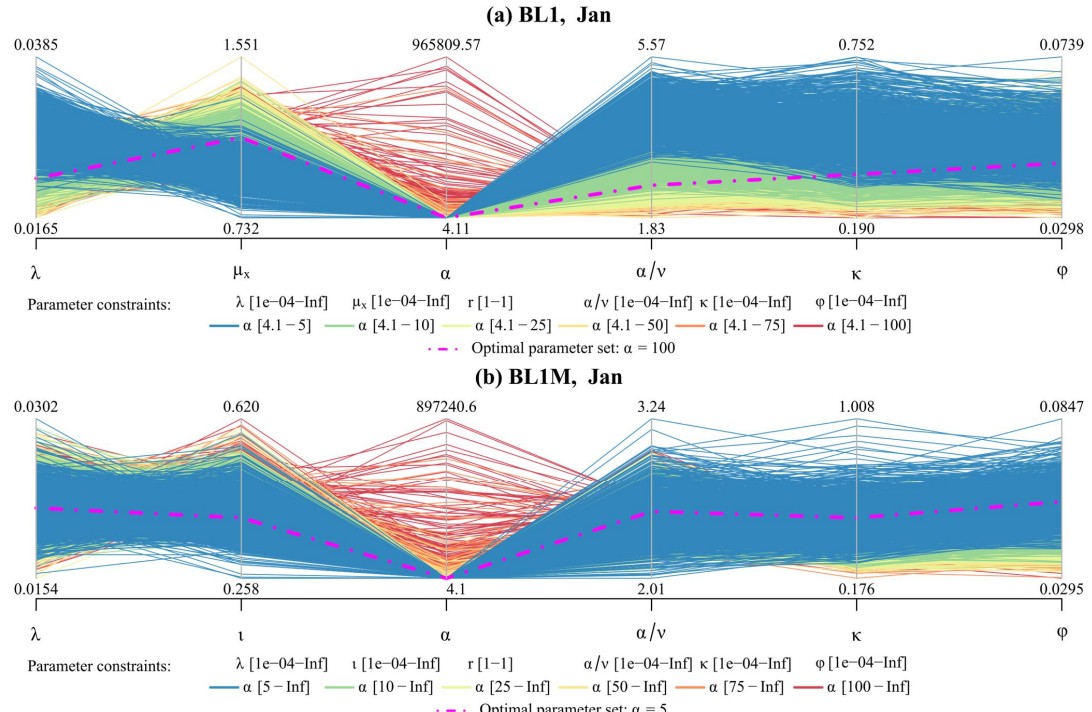

**Figure A1 Parallel coordinate plots for the two randomised Bartlett-Lewis rectangular pulse models, BL1 and BL1M. Plots show the range of January parameter values for uncensored models fitted to Bochum 15 minute rainfall. The dashed magenta lines show the parameter sets corresponding to α = 100 and α = 5 for the BL1 and BL1M models respectively.**

The parallel coordinate plots clearly show the insensitivity of α compared with the other model parameters. When α is

constrained with upper and lower bounds of between 25-50 for the BL1 and BL1M models respectively, α is poorly identified

and can take any value over a very large range (see Fig. A1). When α is constrained with upper and lower bounds of less than

25 for the BL1 and BL1M models respectively, identifiability of α is improved. This insensitivity results from the shape of the

fitted Gamma distribution used to sample η shown in Fig. A2.

As α increases the Gamma distribution converges on the Normal distribution and becomes increasingly flat. Therefore, for

600   high values of α, the probability of randomly sampling anywhere within the distribution is greater compared with low values

of α. For α ≥ 50, the Gamma distribution is approximately normal and the range of η values which may be randomly sampled

by both models is always large resulting in a narrow range of potential Exponential distributions from which to sample 1/L

where L is the cell duration. This impacts the estimation of extremes as shown in Fig. A3. Figure A3 shows extreme rainfall

estimates from the BL1 and BL1M models with α fixed at 5, 50 and 100. For α ≥ 50, extreme rainfall estimation by both

605   models is identical. For α = 5, the BL1 model estimates lower extremes than with higher α values, while the BL1M model





gives improved estimation of the growth curve of extremes. Because of this combination of parameter insensitivity and relative

performance in the extremes, α is fixed at 100 and 5 for the BL1 and BL1M models respectively.

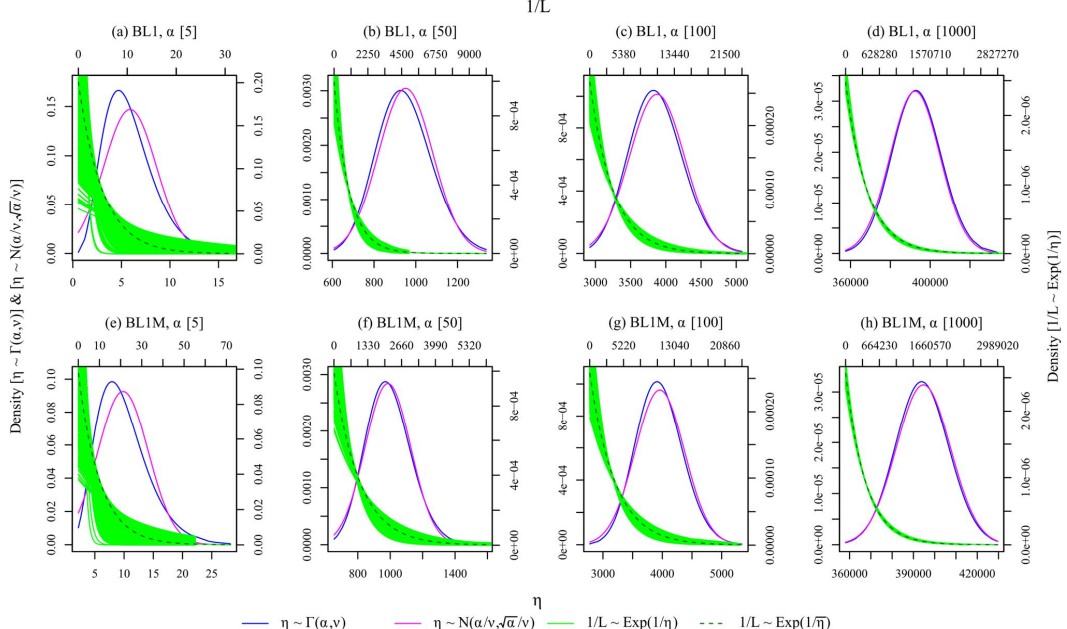

**Figure A2. Fitted Gamma distributions for the cell duration parameter η for the BL1 and BL1M models with α = 5, 50, 100 and 1000. Plots show the equivalent Normal distributions fitted to the mean and standard deviation of the Gamma distributions. The range of Exponential distributions for the cell duration parameter η are obtained by sampling 500 η values from the fitted Gamma distributions. The Exponential distributions for the mean of the fitted Gamma distributions are also shown.**

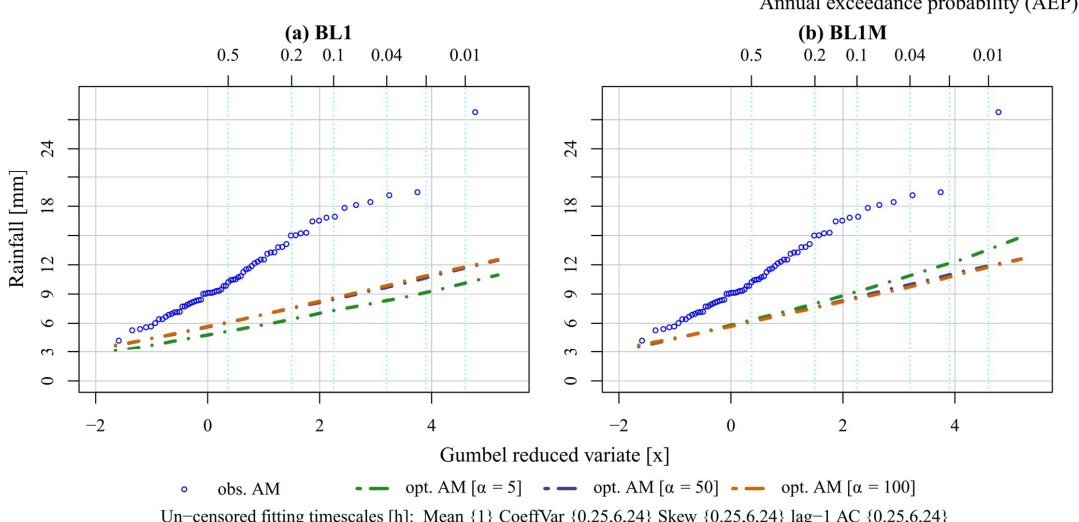

**Figure A3 Sensitivity of extreme value estimation to choice of α for the randomised Bartlett-Lewis models.**



## Appendix B: Fitted model parameters

Tables B1–4 show fitted model parameters for the BL1M model (BLRPR$_X$ in Table 1) for 5 and 15 minute rainfall at Bochum and Atherstone with uncensored and censored data. Censored model parameters correspond to the censors selected in Table 2. Additionally, Tables B1–4 show the objective function value, $S_{min}$, for the fitted parameter set, as well as mechanistic model parameters defined by Wheater et al. (2007b) which are listed below.

Mean number of cells per storm: $\quad \mu_c = 1 + \dfrac{\kappa}{\varphi} \quad$ [-]

Mean cell duration: $\quad \delta_c = \dfrac{\nu}{\alpha - 1} \quad$ [h]

Mean duration of storm activity: $\quad \delta_c = \dfrac{\nu}{(\alpha - 1)\varphi} \quad$ [h]

**Table B1 BL1M model parameters for the Bochum 5 minute data**

| | $\lambda$ [hr$^{-1}$] | $\iota$ [mm] | $\alpha$ [hr] | $\alpha/\nu$ [hr] | $\kappa$ [-] | $\varphi$ [-] | $S_{min}$ [-] | $\mu_c$ [-] | $\delta_c$ [min] | $\delta_s$ [hr] |
|---|---|---|---|---|---|---|---|---|---|---|
| | | | | Uncensored [> 0 mm] | | | | | | |
| Jan | 0.022 | 0.318 | 4.100 | 3.788 | 0.469 | 0.042 | 22.4 | 12.2 | 20.9 | 8.3 |
| Feb | 0.021 | 0.326 | 4.100 | 4.052 | 0.387 | 0.038 | 25.2 | 11.2 | 19.6 | 8.6 |
| Mar | 0.021 | 0.350 | 4.100 | 4.788 | 0.300 | 0.034 | 27.4 | 9.8 | 16.6 | 8.1 |
| Apr | 0.022 | 0.423 | 4.100 | 5.943 | 0.211 | 0.029 | 41.2 | 8.3 | 13.4 | 7.7 |
| May | 0.024 | 0.510 | 4.100 | 7.594 | 0.205 | 0.032 | 48.4 | 7.4 | 10.4 | 5.4 |
| Jun | 0.024 | 0.682 | 4.100 | 9.082 | 0.164 | 0.032 | 55.8 | 6.1 | 8.7 | 4.6 |
| Jul | 0.024 | 0.766 | 4.100 | 9.839 | 0.152 | 0.032 | 57.0 | 5.8 | 8.1 | 4.2 |
| Aug | 0.023 | 0.786 | 4.100 | 9.294 | 0.133 | 0.029 | 54.8 | 5.6 | 8.5 | 4.9 |
| Sep | 0.021 | 0.626 | 4.100 | 7.743 | 0.175 | 0.029 | 46.8 | 7.0 | 10.2 | 5.9 |
| Oct | 0.021 | 0.506 | 4.100 | 6.008 | 0.226 | 0.030 | 32.1 | 8.5 | 13.2 | 7.3 |
| Nov | 0.021 | 0.380 | 4.100 | 4.697 | 0.359 | 0.036 | 28.1 | 11.0 | 16.9 | 7.8 |
| Dec | 0.022 | 0.332 | 4.100 | 3.984 | 0.435 | 0.039 | 29.5 | 12.2 | 19.9 | 8.5 |
| | | | | Censored [> 0.5 mm] | | | | | | |
| Jan | 0.007 | 0.288 | 5.000 | 42.472 | 0.003 | 0.007 | 0.5 | 1.4 | 1.8 | 4.2 |
| Feb | 0.007 | 0.302 | 5.000 | 47.366 | 0.002 | 0.005 | 0.2 | 1.4 | 1.6 | 5.3 |
| Mar | 0.008 | 0.310 | 5.000 | 47.877 | 0.003 | 0.008 | 0.4 | 1.4 | 1.6 | 3.3 |
| Apr | 0.009 | 0.500 | 5.000 | 36.524 | 0.005 | 0.011 | 0.0 | 1.5 | 2.1 | 3.1 |
| May | 0.010 | 0.831 | 5.000 | 27.921 | 0.020 | 0.036 | 0.6 | 1.6 | 2.7 | 1.2 |
| Jun | 0.011 | 1.056 | 5.000 | 21.226 | 0.030 | 0.039 | 1.6 | 1.8 | 3.5 | 1.5 |
| Jul | 0.012 | 1.215 | 5.000 | 21.060 | 0.029 | 0.040 | 0.9 | 1.7 | 3.6 | 1.5 |
| Aug | 0.011 | 1.177 | 5.000 | 21.399 | 0.025 | 0.033 | 0.7 | 1.8 | 3.5 | 1.8 |
| Sep | 0.011 | 1.033 | 5.000 | 26.439 | 0.009 | 0.021 | 0.3 | 1.4 | 2.8 | 2.3 |
| Oct | 0.009 | 0.716 | 5.000 | 31.356 | 0.003 | 0.010 | 0.1 | 1.3 | 2.4 | 4.0 |
| Nov | 0.008 | 0.377 | 5.000 | 39.141 | 0.001 | 0.003 | 0.0 | 1.3 | 1.9 | 10.6 |
| Dec | 0.007 | 0.317 | 5.000 | 41.561 | 0.001 | 0.005 | 0.0 | 1.2 | 1.8 | 6.0 |

**Table B2 BL1M model parameters for the Atherstone 5 minute data**

| | $\lambda$ [hr$^{-1}$] | $\iota$ [mm] | $\alpha$ [hr] | $\alpha/\nu$ [hr] | $\kappa$ [-] | $\varphi$ [-] | $S_{min}$ [-] | $\mu_c$ [-] | $\delta_c$ [min] | $\delta_s$ [hr] |
|---|---|---|---|---|---|---|---|---|---|---|
| | | | | Uncensored [> 0 mm] | | | | | | |
| Jan | 0.023 | 0.095 | 4.100 | 79.758 | 0.157 | 0.005 | 49.4 | 32.4 | 1.0 | 3.3 |
| Feb | 0.022 | 0.083 | 4.100 | 110.403 | 0.117 | 0.004 | 54.4 | 30.3 | 0.7 | 3.0 |
| Mar | 0.022 | 0.097 | 4.100 | 64.565 | 0.139 | 0.005 | 61.7 | 28.8 | 1.2 | 4.1 |
| Apr | 0.020 | 0.137 | 4.100 | 46.008 | 0.161 | 0.007 | 41.1 | 24.0 | 1.7 | 4.1 |
| May | 0.018 | 0.233 | 4.100 | 28.827 | 0.172 | 0.011 | 28.3 | 16.6 | 2.8 | 4.2 |
| Jun | 0.017 | 0.328 | 4.100 | 22.831 | 0.195 | 0.016 | 20.5 | 13.2 | 3.5 | 3.6 |
| Jul | 0.017 | 0.395 | 4.400 | 19.700 | 0.186 | 0.018 | 20.0 | 11.3 | 3.9 | 3.6 |
| Aug | 0.017 | 0.338 | 4.400 | 22.385 | 0.209 | 0.018 | 21.5 | 12.6 | 3.5 | 3.2 |
| Sep | 0.018 | 0.253 | 4.100 | 26.711 | 0.224 | 0.014 | 28.2 | 17.0 | 3.0 | 3.5 |





| | | | | | | | | | |
|---|---|---|---|---|---|---|---|---|---|
| Oct | 0.019 | 0.164 | 4.100 | 37.406 | 0.245 | 0.010 | 37.6 | 25.5 | 2.1 | 3.5 |
| Nov | 0.021 | 0.115 | 4.100 | 52.659 | 0.234 | 0.007 | 47.6 | 34.4 | 1.5 | 3.6 |
| Dec | 0.022 | 0.091 | 4.100 | 84.400 | 0.185 | 0.005 | 49.1 | 38.0 | 0.9 | 3.1 |

|  | Censored [> 0.6 mm] | | | | | | | | |
|---|---|---|---|---|---|---|---|---|---|
| Jan | 0.007 | 0.316 | 5.000 | 43.290 | 0.029 | 0.057 | 5.5 | 1.5 | 1.7 | 0.5 |
| Feb | 0.007 | 0.250 | 5.000 | 52.351 | 0.025 | 0.045 | 6.7 | 1.6 | 1.4 | 0.5 |
| Mar | 0.007 | 0.275 | 5.000 | 56.244 | 0.014 | 0.028 | 4.5 | 1.5 | 1.3 | 0.8 |
| Apr | 0.007 | 0.392 | 5.000 | 50.130 | 0.012 | 0.020 | 1.9 | 1.6 | 1.5 | 1.2 |
| May | 0.007 | 0.594 | 5.000 | 37.073 | 0.014 | 0.022 | 0.9 | 1.6 | 2.0 | 1.5 |
| Jun | 0.008 | 0.695 | 5.000 | 31.026 | 0.029 | 0.036 | 1.9 | 1.8 | 2.4 | 1.1 |
| Jul | 0.008 | 0.805 | 5.000 | 26.653 | 0.027 | 0.034 | 0.6 | 1.8 | 2.8 | 1.4 |
| Aug | 0.008 | 0.719 | 5.000 | 29.868 | 0.027 | 0.032 | 0.5 | 1.8 | 2.5 | 1.3 |
| Sep | 0.008 | 0.644 | 5.000 | 33.789 | 0.014 | 0.023 | 1.5 | 1.6 | 2.2 | 1.6 |
| Oct | 0.008 | 0.463 | 5.000 | 46.623 | 0.009 | 0.018 | 0.2 | 1.5 | 1.6 | 1.5 |
| Nov | 0.008 | 0.369 | 5.000 | 46.777 | 0.007 | 0.021 | 1.6 | 1.3 | 1.6 | 1.3 |
| Dec | 0.007 | 0.318 | 5.000 | 49.550 | 0.015 | 0.037 | 5.2 | 1.4 | 1.5 | 0.7 |

620 **Table B3 BL1M model parameters for the Bochum 15 minute data**

| | $\lambda$ [hr$^{-1}$] | $\iota$ [mm] | $\alpha$ [hr] | $\alpha/\nu$ [hr] | $\kappa$ [-] | $\varphi$ [-] | $S_{min}$ [-] | $\mu_c$ [-] | $\delta_c$ [min] | $\delta_s$ [hr] |
|---|---|---|---|---|---|---|---|---|---|---|
| | Uncensored [> 0 mm] | | | | | | | | | |
| Jan | 0.022 | 0.373 | 4.100 | 2.562 | 0.545 | 0.059 | 18.1 | 10.2 | 31.0 | 8.7 |
| Feb | 0.021 | 0.375 | 4.100 | 2.752 | 0.458 | 0.052 | 20.7 | 9.8 | 28.8 | 9.2 |
| Mar | 0.021 | 0.376 | 4.100 | 3.251 | 0.404 | 0.049 | 23.6 | 9.2 | 24.4 | 8.3 |
| Apr | 0.022 | 0.448 | 4.100 | 4.023 | 0.292 | 0.043 | 34.4 | 7.8 | 19.7 | 7.6 |
| May | 0.024 | 0.518 | 4.100 | 5.323 | 0.307 | 0.049 | 40.3 | 7.3 | 14.9 | 5.1 |
| Jun | 0.024 | 0.665 | 4.100 | 6.799 | 0.254 | 0.048 | 46.7 | 6.3 | 11.7 | 4.1 |
| Jul | 0.024 | 0.738 | 4.100 | 7.496 | 0.241 | 0.048 | 46.8 | 6.0 | 10.6 | 3.7 |
| Aug | 0.023 | 0.749 | 4.100 | 7.054 | 0.219 | 0.044 | 43.5 | 6.0 | 11.2 | 4.3 |
| Sep | 0.021 | 0.624 | 4.100 | 5.539 | 0.268 | 0.045 | 35.0 | 7.0 | 14.3 | 5.3 |
| Oct | 0.021 | 0.529 | 4.100 | 4.079 | 0.321 | 0.045 | 23.7 | 8.1 | 19.5 | 7.2 |
| Nov | 0.021 | 0.434 | 4.100 | 3.117 | 0.439 | 0.051 | 19.3 | 9.6 | 25.5 | 8.3 |
| Dec | 0.022 | 0.411 | 4.100 | 2.593 | 0.463 | 0.052 | 20.1 | 9.9 | 30.6 | 9.8 |
| | Censored [> 1.0 mm] | | | | | | | | | |
| Jan | 0.008 | 0.340 | 5.000 | 21.917 | 0.005 | 0.010 | 0.4 | 1.5 | 3.4 | 5.7 |
| Feb | 0.008 | 0.355 | 5.000 | 23.830 | 0.004 | 0.010 | 0.3 | 1.4 | 3.1 | 5.2 |
| Mar | 0.008 | 0.401 | 5.000 | 22.721 | 0.005 | 0.014 | 0.7 | 1.4 | 3.3 | 3.9 |
| Apr | 0.009 | 0.629 | 5.000 | 18.092 | 0.007 | 0.016 | 0.6 | 1.4 | 4.1 | 4.3 |
| May | 0.010 | 0.987 | 5.000 | 15.213 | 0.014 | 0.026 | 1.8 | 1.5 | 4.9 | 3.2 |
| Jun | 0.012 | 1.240 | 5.000 | 13.109 | 0.017 | 0.026 | 1.5 | 1.7 | 5.7 | 3.7 |
| Jul | 0.012 | 1.397 | 5.000 | 13.518 | 0.017 | 0.025 | 1.2 | 1.7 | 5.5 | 3.7 |
| Aug | 0.012 | 1.372 | 5.000 | 13.857 | 0.013 | 0.020 | 0.4 | 1.7 | 5.4 | 4.5 |
| Sep | 0.010 | 1.141 | 5.000 | 15.219 | 0.010 | 0.019 | 0.0 | 1.5 | 4.9 | 4.3 |
| Oct | 0.009 | 0.809 | 5.000 | 16.864 | 0.005 | 0.011 | 0.0 | 1.5 | 4.4 | 6.7 |
| Nov | 0.008 | 0.442 | 5.000 | 18.891 | 0.004 | 0.008 | 0.1 | 1.5 | 4.0 | 8.3 |
| Dec | 0.007 | 0.385 | 5.000 | 21.324 | 0.003 | 0.007 | 0.1 | 1.4 | 3.5 | 8.4 |

**Table B4 BL1M model parameters for the Atherstone 15 minute data**

| | $\lambda$ [hr$^{-1}$] | $\iota$ [mm] | $\alpha$ [hr] | $\alpha/\nu$ [hr] | $\kappa$ [-] | $\varphi$ [-] | $S_{min}$ [-] | $\mu_c$ [-] | $\delta_c$ [min] | $\delta_s$ [hr] |
|---|---|---|---|---|---|---|---|---|---|---|
| | Uncensored [> 0 mm] | | | | | | | | | |
| Jan | 0.022 | 0.147 | 4.100 | 14.734 | 0.505 | 0.025 | 17.6 | 21.2 | 5.4 | 3.6 |
| Feb | 0.022 | 0.129 | 4.100 | 17.115 | 0.445 | 0.021 | 18.7 | 22.2 | 4.6 | 3.7 |
| Mar | 0.022 | 0.141 | 4.100 | 16.373 | 0.354 | 0.019 | 24.5 | 19.6 | 4.8 | 4.3 |
| Apr | 0.020 | 0.184 | 4.100 | 14.038 | 0.373 | 0.022 | 20.0 | 18.0 | 5.7 | 4.3 |
| May | 0.018 | 0.302 | 4.100 | 10.715 | 0.327 | 0.027 | 19.1 | 13.1 | 7.4 | 4.6 |
| Jun | 0.017 | 0.410 | 4.100 | 9.692 | 0.335 | 0.035 | 19.4 | 10.6 | 8.2 | 3.9 |
| Jul | 0.017 | 0.482 | 4.100 | 9.013 | 0.304 | 0.038 | 21.9 | 9.0 | 8.8 | 3.9 |
| Aug | 0.018 | 0.408 | 4.100 | 9.896 | 0.366 | 0.040 | 22.4 | 10.2 | 8.0 | 3.3 |
| Sep | 0.019 | 0.335 | 4.100 | 9.388 | 0.440 | 0.039 | 24.0 | 12.3 | 8.5 | 3.6 |
| Oct | 0.019 | 0.243 | 4.100 | 9.798 | 0.569 | 0.036 | 22.6 | 16.8 | 8.1 | 3.7 |
| Nov | 0.021 | 0.199 | 4.100 | 10.078 | 0.620 | 0.034 | 20.4 | 19.2 | 7.9 | 3.9 |





| | | | | | | | | | | |
|---|---|---|---|---|---|---|---|---|---|---|
| Dec | 0.021 | 0.164 | 4.100 | 11.699 | 0.640 | 0.031 | 15.9 | 21.6 | 6.8 | 3.6 |
| Censored [> 0.6 mm] | | | | | | | | | | |
| Jan | 0.010 | 0.472 | 5.000 | 12.186 | 0.047 | 0.048 | 0.5 | 2.0 | 6.2 | 2.1 |
| Feb | 0.010 | 0.400 | 5.000 | 13.782 | 0.041 | 0.046 | 0.9 | 1.9 | 5.4 | 2.0 |
| Mar | 0.010 | 0.399 | 5.000 | 15.383 | 0.029 | 0.038 | 0.3 | 1.8 | 4.9 | 2.1 |
| Apr | 0.010 | 0.501 | 5.000 | 13.827 | 0.046 | 0.039 | 0.2 | 2.2 | 5.4 | 2.3 |
| May | 0.010 | 0.780 | 5.000 | 11.245 | 0.031 | 0.028 | 0.1 | 2.1 | 6.7 | 4.0 |
| Jun | 0.009 | 0.904 | 5.000 | 11.080 | 0.058 | 0.039 | 0.0 | 2.5 | 6.8 | 2.9 |
| Jul | 0.010 | 1.056 | 5.000 | 10.606 | 0.041 | 0.033 | 0.1 | 2.2 | 7.1 | 3.6 |
| Aug | 0.011 | 1.082 | 5.000 | 9.943 | 0.025 | 0.024 | 0.2 | 2.0 | 7.5 | 5.2 |
| Sep | 0.011 | 0.924 | 5.000 | 9.330 | 0.019 | 0.017 | 0.2 | 2.1 | 8.0 | 7.9 |
| Oct | 0.010 | 0.711 | 5.000 | 8.875 | 0.029 | 0.024 | 0.3 | 2.2 | 8.5 | 5.9 |
| Nov | 0.010 | 0.522 | 5.000 | 9.761 | 0.057 | 0.040 | 0.5 | 2.4 | 7.7 | 3.2 |
| Dec | 0.010 | 0.484 | 5.000 | 10.229 | 0.064 | 0.048 | 0.8 | 2.3 | 7.3 | 2.5 |

**Author contribution**

David Cross designed the experiments, carried them out and prepared the manuscript. Christian Onof, Hugo Winter and Pietro Bernardara supervised the work and reviewed the manuscript preparation.

**Competing interests**

The authors declare that they have no conflict of interest.

**Acknowledgements**

David Cross is grateful for the award of an Industrial Case Studentship from the Engineering and Physical Sciences Research Council and EDF Energy. The Environment Agency of England are gratefully acknowledged for providing the UK rainfall data, and Deutsche Montan Technologie and Emschergenossenschaft / Lippeverband in Germany are gratefully acknowledged for providing the Bochum data.

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
