# Peer review of "Censored rainfall modelling for estimation of fine-scale extremes"

_Hydrology and Earth System Sciences, 2017_

## Referee Comment (RC1) · Anonymous Referee #1 · 25 Sep 2017

**General comments**

The aim of this paper is to improve the ability of mechanistic rainfall models extremes to reproduce rainfall extremes. This is achieved by fitting these models to the amounts by which rainfall totals exceed a certain threshold level, or *censor*. Totals below this level are censored: their value is taken to be zero when the models are fitted. Applying a threshold to data is a standard approach in extreme value modelling. It's use here is a novel and interesting idea. If modelling rainfall extremes is the primary goal then using the censor to reduce the influence of small totals is sensible.

Of course, a key issue is the choice of censor: a low censor may not achieve the desired objective but as the censor is raised the precision of estimation reduces and,

in the current context, may exacerbate the parameter identifiability problems to which the fitting of these models are prone. This is analogous to the choice of threshold in an extreme value analysis and therefore it is unsurprising that in Figure 13 they consider informing this choice using a graphical approach that is common in extreme value modelling. It may be productive to explore other methods proposed in the extreme value threshold selection literature.

Overall I am positive about this paper. My main criticism is that better reproduction of rainfall extremes is achieved by tuning various things: censor; model parameterisation; fitting properties; perhaps even the model itself, in order to achieve this objective. There is further work to be done to provide methodology to make these choices.

**Specific comments**

**page 11, line 288**. This isn't quite correct. These weights are not optimal. However, in practice they are close to being optimal and are easier to estimate than the weights that are optimal. On that note: how are these weights estimated?

**page 12, lines 305-308**. This isn't quite correct. The sampling distribution of the GMM estimators is approximated by a MVN distribution, i.e. there is an approximation involved and the result is for the rule that is used to calculate the estimates, rather than the estimates themselves. Line 306: Hessian of what? Lines 307-308. This sentence isn't clear. Presumably the point is that the calculation of confidence intervals can fail in some cases. Perhaps it would be sufficient to reverse the ordering of the sentence to make the causation clearer.

**page 13, line 345**. If you are interested in the 1000 year return level why not simulate 100 realisations of 1000 years duration?

**pages 14-16, Figures 4-6**. There seems to be a slight upward curvature in the lines

based on the 10000 year simulation. In the context of an extreme value analysis this is consistent with the shape parameter discussed on page 4 (line 111) being positive. This might be worth a brief comment. Is there any work that examines how the extreme value properties of this type of model relate to the model parameters and therefore provide a link to the theoretical basis that underpins extrapolation from extreme value models?

**page 17, lines 385-386**. I disagree. I suppose that it depends what you mean by "poorly identified". However, it is to be expected that as the censor is increased uncertainty about model parameters increases. If we think that we need a larger censor, because otherwise there is systematic underestimation of extreme rainfall totals, then we need to accept higher levels of parameter uncertainty.

**page 18, Section 6.2**. Is this level of tinkering with the choice of censor justified? Having a different censor for different levels of data aggregation feels like cherry-picking. Also, in the previous section an argument was made against a censor of 0.6mm for Atherstone but now it is being used.

**page 19, Figure 8 caption (and elsewhere)**. "optimal censors" seems like a bold claim given the difficultly of choosing the censor.

**page 28, line 516**. The independence criterion *isn't* a requirement in extreme value modelling. See Fawcett and Walshaw (2012) Estimating return levels from serially dependent extremes. Environmetrics 23(3), 272-283.

**Technical corrections**

**page 3, line 81**. At this point, or perhaps even in the abstract, it is worth explaining briefly the nature of the censoring. At the moment we need to wait until page 7 for this.

**page 4, line 118**. "behavioural parameterizations". Given that you use this term later it

would be worth explaining (somewhere) what this means in the context of the current paper.

**page 6, line 160**. "Lower variability" may be better than "less variance".

**page 10, line 262**. Presumably the reason for the missing data in 1974-75 was political rather than environmental. It might be worth noting that the fact that the data are missing is not expected to be informative about rainfall totals.

**page 12, line 313**. "ldots extreme values continued to be underestimated . . . " might be better.

**pages 14-16, Figures 4-6**. The plots would be clearer if the scale on the lower horizontal axis was return level in years. The AEP on the upper horizontal axis would then be unnecessary. The scale of the Gumbel reduced variate adds no information in itself. These plots are quite crowded and

**page 17, line 358**. I'm not sure that I would use "confidence intervals" here. Perhaps "simulation bands"? ... and say explicitly what this means, i.e. how the lines in the plot are calculated.

**page 17, line 379**. I'm not sure what this sentence means. Are we supposed to be looking at Figure 7 for evidence of this?

**pages 20-21, Sections 6.2.1, 6.2.2 and 6.2.3**. I don't see the point of including these sections. Section 6.2.1 shows exactly what we expect: by excluding properties that are difficult to reproduce we are able to reproduce well the properties that are not excluded. The comparison in Section 6.2.2 is unsatisfactory because we cannot compare like with like, owing to the truncation of the data but not the model. Section 6.2.3 just shows that there are clear local minima in the objective function but we can't expect to search too far in the search for confidence limits.

**page 27, Figure 14**. I don't think that these figures add much to the statistics concerning the proportions of totals lying below the censors, with the possible exception of the

visualisation of the resolution of the Atherstone data.

**page 28, line 514**. The rule to try to create independent peaks needs to be given earlier: before the concept appears in Table 3.

**pages 28-30**. Do we need both "Further discussion" and "Conclusions"?

**page 31, line 591**. Is the first inequality sign the wrong way round?

**page 32, Figure A.2**. Below and to the right of the plot is says that $1/L$ has an exponential distribution, which, according to the description of the models on page 8, isn't true.

---

## Referee Comment (RC2) · Anonymous Referee #2 · 9 Oct 2017

General comments.

The research work presented in the manuscript develops a new methodology to estimate fine-scale rainfall extremes. Although there has been a substantial amount of work done, by many authors over the years, on stochastic point process models for rainfall, most of the models proposed tend to underestimate the rainfall extremes at fine-scales. Estimation or reproduction of extreme rainfall at hourly and sub-hourly scales is a well-known problem. In this context, this paper attempts to address this problem by using a censored approach to model rainfall extremes. This is in a way similar to the Excess Over Threshold (EOT) method commonly used in extreme value modelling, but here a stochastic mechanistic model is used along with this idea. Application of this novel idea of censured modelling approach is illustrated in the estimation

of fine-scale rainfall extremes from two different regions to provide an improved representation of extremes.

The paper gives an excellent coverage of the history of work carried out in this area to convey the rationale for the need to study or explore alternative methods for fine-scale extremes.

The success of this new approach, of course depends heavily on the choice of the censor level and, hence, emphasis was placed on finding appropriate value of the censor for the application. If the estimation of extreme rainfall is the main objective of the study then using this censored approach is certainly a useful tool and worthwhile addition to the existing methods.

One drawback in the proposed approach might be the amount of fine-tuning required to get the best set of potential estimates for the extreme rainfall with respect to model, its parameterisation, censor, statistics used in fitting as well as aggregation levels. This level of tweaking or fine-tuning might prove to be a lot to generate sufficient interest amongst practitioners. The rationale behind the need to make these choices, however, has been explained in the manuscript though. Specific comments.

Line 257: Would be useful to give a reason for the assumption of rain cells starting at the storm origin.

Line 316-319: Can appreciate the reason given for the choice of fitting statistics used for model calibration, but the question now is that how do the parameter estimates compare when the same fitting stats are used for uncensored fitting? Has this been explored?

Line 358: Perhaps you need to explain what you meant by behavioural parameters for the readers.

Line 358: 95% confidence intervals: unless you are using the standard errors of the estimates, I am not sure whether "confidence" interval is the appropriate terminology

here. Simulation bands?

Line 396: Not sure why your validation for Atherstone was based on 0.6mm censor which seem to contradict your statement on lines 375-380. Some insight/explanation would be useful to the readers.

Line 396: Table 2. Different censor for different sites is understandable. However, why do you need to use the same sensor at 3 different aggregation level for Atherstone while using different censors for the 3 levels of aggregation for Bochum?

Fig 8: row 2. The nice seasonal pattern observed in the mean rainfall for Atherstone at 5 and 15 minutes has become less prominent or disappeared at 60 minutes. Can you comment on why? No observation or comment was made about this.

---

## Author Response (AR2)

**Author's response**

Review of HESS discussion paper reference no. hess-2017-437

**Censored rainfall modelling for estimation of fine–scale extremes**

David Cross, Christian Onof, Hugo Winter, and Pietro Bernardara

We are very grateful to two anonymous referees and the handling editor for their constructive comments and suggestions on our manuscript. We set out here our responses to those and list our changes to the manuscript.

Section 1. point-by-point response to the reviewers' and editor's comments.

Section 2. list of all relevant changes in the manuscript.

10 Section 3. marked-up manuscript version showing track changes.

**1. Responses to reviewers' and editor's comments**

**1.1 Anonymous Referee #1**

15 **General comments**

The aim of this paper is to improve the ability of mechanistic rainfall models extremes to reproduce rainfall extremes. This is achieved by fitting these models to the amounts by which rainfall totals exceed a certain threshold level, or censor. Totals below this level are censored: their value is taken to be zero when the models are fitted. Applying a threshold to data is a standard approach in extreme value modelling. It's use here is a novel 20 and interesting idea. If modelling rainfall extremes is the primary goal then using the censor to reduce the influence of small totals is sensible. Discussion paper Of course, a key issue is the choice of censor: a low censor may not achieve the desired objective but as the censor is raised the precision of estimation reduces and, in the current context, may exacerbate the parameter identifiability problems to which the fitting of these models are prone. This is analogous to the choice of threshold in an extreme value analysis and therefore it is unsurprising 25 that in Figure 13 they consider informing this choice using a graphical approach that is common in extreme value Interactive modelling. It may be productive to explore other methods proposed in the extreme comment value threshold selection literature.

Overall I am positive about this paper. My main criticism is that better reproduction of rainfall extremes is achieved by tuning various things: censor; model parameterisation; fitting properties; perhaps even the model 30 itself, in order to achieve this objective. There is further work to be done to provide methodology to make these choices.

**Specific comments**

**page 11, line 288.** This isn't quite correct. These weights are not optimal. However, in practice they are close to being optimal and are easier to estimate than the weights that are optimal. On that note: how are these weights 35 estimated?

The estimation of weights follows the theory and procedure set out in the MOMFIT manual (Chandler et al. 2010). First, the covariance matrix of mean summary statistics $\Sigma$ is estimated using Equation 1.

$$\hat{\Sigma} = \frac{1}{n(n-1)} \sum_{i=1}^{n} (T_i - \bar{T})(T_i - \bar{T})'$$

*Equation 1*

$T_i$ is the vector of summary statistics for the $i$th period. $\bar{T}$ is the vector of mean summary statistics and n is the total number of periods which in practice is the length of the dataset in years. The optimal choice of weights W would be closely approximated by the inverse of the covariance matrix given in Equation 2.

$$W = \Sigma^{-1}$$

*Equation 2*

However, as reported by Chandler et al. (2010), this results in highly inaccurate standard errors because the unique elements of Σ are estimated separately by Equation 1. To overcome this, a reasonable approximation for the weights is to take the inverse of the observed variance where $t_i$ is the vector of diagonal elements of Σ.

$$\omega_i = 1/var(t_i)$$

We can update the manuscript to indicate that the estimation of weights is not optimal, but provides a robust estimate.

**page 12, lines 305-308.** This isn't quite correct. The sampling distribution of the GMM estimators is approximated by a MVN distribution, i.e. there is an approximation involved and the result is for the rule that is used to calculate the estimates, rather than the estimates themselves. Line 306: Hessian of what? Lines 307-308. This sentence isn't clear. Presumably the point is that the calculation of confidence intervals can fail in some cases. Perhaps it would be sufficient to reverse the ordering of the sentence to make the causation clearer.

Line 305: We can amend the text to highlight that the distribution of the estimator resulting from the minimisation routine is approximately multivariate normal.

Line 306: The covariance matrix is estimated with the Hessian of the least squared objective function S given on line 280. We can revise the text to make this clearer.

Lines 307-308: Indeed, the sentence was provided to highlight that the calculation of confidence intervals can fail. We will reverse the sentence as below.

*On occasions that the model parameters are poorly identified, it may not be possible to calculate the Hessian of the objective function preventing the estimation of parameter uncertainty.*

**page 13, line 345.** If you are interested in the 1000 year return level why not simulate 100 realisations of 1000 years duration?

Extreme value estimation up to the 1000-year return level is provided to indicate the potential magnitude of rarer events. Estimation up to the 1000-year return level is performed for the optimum parameter set, therefore there is no sampling from the MVN distribution of model parameters. To ensure these estimates are reasonably stable up to 1000 years, simulations have been extended to 10k years. Furthermore, simulation bands for 100 simulations of 1000 years would differ from the 100 year simulation bands shown and make the already busy figures more crowded.

For validation that the observed data at each site are sampled from the sampling distribution, it makes sense to derive simulation bands for a length of data that is of the same order of magnitude as that of the observations. Therefore, the duration of 100 years was chosen to cover the range of the data at both sites and to enable comparison between sites.

**pages 14-16, Figures 4-6.** There seems to be a slight upward curvature in the lines based on the 10000 year simulation. In the context of an extreme value analysis this is consistent with the shape parameter discussed on page 4 (line 111) being positive. This might be worth a brief comment. Is there any work that examines how the extreme value properties of this type of model relate to the model parameters and therefore provide a link to the theoretical basis that underpins extrapolation from extreme value models?

This is a good suggestion and we will comment on this as suggested in the discussion. That said, we are not aware of any such work, but we will briefly review the literature again to investigate.

**page 17, lines 385-386.** I disagree. I suppose that it depends what you mean by "poorly identified". However, it is to be expected that as the censor is increased un- certainty about model parameters increases. If we think that we need a larger censor, because otherwise there is systematic underestimation of extreme rainfall totals, then we need to accept higher levels of parameter uncertainty.

This is a good point, although we have observed the same with uncensored models. Overall, our choice of summary statistics gives rise to apparently very well identified model parameters at sub-hourly scales. The comments here relate specifically to estimation at the hourly scale.

Our statement that the parameters are poorly identified may be overly strong. We can change this to state that confidence in the estimation is reducing.

**page 18, Section 6.2.** Is this level of tinkering with the choice of censor justified? Having a different censor for different levels of data aggregation feels like cherry-picking. Also, in the previous section an argument was made against a censor of 0.6mm for Atherstone but now it is being used.

The model is fitted separately for each temporal resolution which explains why we have different censors. Because of the effect of aggregation, we cannot use a model censored at one resolution to estimate rainfall extremes at a coarser one. Therefore, censored model parameters are scale dependent which is explained in section 3. Furthermore, we believe that the use of different censors at different levels of aggregation is justified on the basis that the distribution of rainfall amounts differs between aggregation levels.
The research as presented is exploratory and intended to investigate the potential for censoring in estimating extremes. In this context, a range of censors have been investigated and shown to be effective at improving the extreme value estimation of the models, and a choice had to be made for validation. Considering that, we agree that it would have been more consistent to select 0.2 mm for validation of the Atherstone hourly censored model and we will make that change.

**page 19, Figure 8 caption (and elsewhere).** "optimal censors" seems like a bold claim given the dificultly of choosing the censor.

We agree and will refer instead to selected censors.

**page 28, line 516.** The independence criterion isn't a requirement in extreme value modelling. See Fawcett and Walshaw (2012) Estimating return levels from serially dependent extremes. Environmetrics 23(3), 272-283.

Thank you for the reference, we were unaware of this research and appreciate its relevance in the context of ours.

We note that the methods set out in Fawcett and Walshaw (2012) requires the estimation of the extremal index which appears to be subjective and potentially non-trivial. Therefore, we feel that our comparison with the standard peaks over threshold approach in which independent cluster peaks are identified is still valid. We will highlight this in the discussion with reference to this research.

**Technical corrections**

**page 3, line 81.** At this point, or perhaps even in the abstract, it is worth explaining briefly the nature of the censoring. At the moment we need to wait until page 7 for this.

We will make the following change to the text.

*To test our hypothesis, a simple approach is proposed in which low observations for fine–scale data are censored from the models in calibration. For a given temporal resolution, a censor amount is set. Rainfall below the censor is set to zero and rainfall over the censor is reduced by the censor amount.*

**page 4, line 118. "behavioural parameterizations".** Given that you use this term later it would be worth explaining (somewhere) what this means in the context of the current paper.

This was also highlighted by Anonymous Referee #2 therefore the response below is the same as that provided to referee #2.

We have used the term "behavioural parameters" by analogy with Beven and Binley (1992). We have used the term to refer to well identified models. We have found that for well identified parameters with narrow 95% confidence intervals, simulation bands on the extreme value estimates are correspondingly narrow. As the parameters become less well identified, their 95% confidence intervals increase giving rise to extreme value estimates which deviate significantly from the observations, which in turn results in significant deviation of the simulation band upper limit. This effect is shown in Fig.11 resulting from the very large parameter uncertainty shown in Fig.12.

We will remove the reference to "behavioural parameterizations" in the context of this research and change all references to well identified parameters.

**page 6, line 160.** "Lower variability" may be better than "less variance".

Agreed. We will make this change in the manuscript.

**page 10, line 262.** Presumably the reason for the missing data in 1974-75 was political rather than environmental. It might be worth noting that the fact that the data are missing is not expected to be informative about rainfall totals.

We do not know why the data are missing, but we can certainly highlight that this is not expected to affect the results. We will make the following change to the text.

*Atherstone is a tipping bucket rain gauge (TBR) operated and maintained by the Environment Agency of England. The record duration is 48 years from 1967 to 2015, with one notable period of missing data from January 1974 to March 1975. The reason for the missing data is unknown, although it is not expected to affect model fitting and the estimation of extremes.*

**page 12, line 313.** "ldots extreme values continued to be underestimated . . . " might be better.

We will change this sentence to the following.

*While good model fits were obtained for some low censors, extreme value estimation continued to be underestimated.*

**pages 14-16, Figures 4-6**. The plots would be clearer if the scale on the lower horizontal axis was return level in years. The AEP on the upper horizontal axis would then be unnecessary. The scale of the Gumbel reduced variate adds no information in itself. These plots are quite crowded and

We agree that the Gumbel reduced variate adds no additional information given that the AEP is provided on the secondary x axis. We also agree that removing this and moving the AEP to the primary x axis will simplify the

plots. However, we propose to keep the content of the plots unchanged as they show the convergence in estimation for all AEPs up to 0.001 for increasing censors.

**page 17, line 358.** I'm not sure that I would use "confidence intervals" here. Perhaps "simulation bands"? ... and say explicitly what this means, i.e. how the lines in the plot are calculated.

This was also highlighted by Anonymous Referee #2 therefore the response below is the same as that provided to referee #2. We agree with the suggestion and will change all occurrences in the manuscript.

There are in fact two issues here: if we were doing very long simulations with practically no random noise (so that another simulation would yield practically the same result), then we would have identified approximate confidence intervals. But with the shorter simulation length, both parameter uncertainty and the randomness of the model are combined in the spread we observe in the simulated statistics, so that 'simulation bands' is indeed a better descriptor.

**page 17, line 379.** I'm not sure what this sentence means. Are we supposed to be looking at Figure 7 for evidence of this?

The reader should be looking at Fig.6 for evidence of this divergence in estimation. This is explained with the aid of Fig.7. We will make the following change to the text.

*The mean of the MVN realisations for the BL1M model at Atherstone with the 0.6 and 0.8 mm censors (see Fig.6) diverges from the optimum because of the generation of unrealistic extremes. This divergence is also observable in the larger spread of 95% simulation intervals over 100 realisations.*

**pages 20-21, Sections 6.2.1, 6.2.2 and 6.2.3.** I don't see the point of including these sections. Section 6.2.1 shows exactly what we expect: by excluding properties that are difficult to reproduce we are able to reproduce well the properties that are not excluded. The comparison in Section 6.2.2 is unsatisfactory because we cannot compare like with like, owing to the truncation of the data but not the model. Section 6.2.3 just shows that there are clear local minima in the objective function but we can't expect to search too far in the search for confidence limits.

It is not always the case that the fitted parameters well reproduce the summary statistics used in fitting. The purpose of these checks is to ensure they do given that the data are censored. That said, given that the ability of the models to reproduce the summary statistics used in fitting at both sites is equally good, we could reduce this section by only showing plots for one site. We could then state that comparable performance is achieved at both sites.

We take the point about checking skewness. Given that we've already highlighted that skewness is not expected to be well reproduced because of the truncation of the data we are happy to remove these plots. However, we feel there is still validity in checking the proportion of dry periods as this property is strongly affected by removing low observation.

We would be happy to remove the profile objective function plots given that there is other evidence of good parameter identifiability with high confidence on the parameter estimates.

**page 27, Figure 14.** I don't think that these figures add much to the statistics concerning the proportions of totals lying below the censors, with the possible exception of the visualisation of the resolution of the Atherstone data.

These figures were included to demonstrate graphically how much data is removed by censoring. Given that the methodology for selecting a censor presented in this research is based on a graphical approach, these figures are useful to understand the rainfall quantiles which have given rise to well parameterised censored models. Until an

alternative method is developed to optimise the censor, we feel these plots will aid other practitioners in estimating rainfall extremes with censoring. Therefore, we propose to keep these plots in the manuscript.

**page 28, line 514**. The rule to try to create independent peaks needs to be given earlier: before the concept appears in Table 3.

Noted. We will bring this forward so that it is highlighted before Table 3.

**pages 28-30**. Do we need both "Further discussion" and "Conclusions"?

We can look at combining these into one section possibly called *Further discussion and conclusions* or just *Conclusions*.

**page 31, line 591**. Is the first inequality sign the wrong way round?

Yes, it is. Thank you for highlighting this. We will correct this in the manuscript.

**page 32, Figure A.2**. Below and to the right of the plot is says that 1/L has an exponential distribution, which, according to the description of the models on page 8, isn't true.

Thank you for highlighting this inconsistency. We will correct this and update Fig.A.2. We also notice that the parameterizations for X and L are listed wrongly in line 225. We will revise this as follows.

*Both X and L are assumed to be independent of each other and follow exponential distributions with parameters $1/\mu_x$ and η respectively.*

**References**

Beven, K. and Binley, A.: The future of distributed models: Model calibration and uncertainty prediction, Hydrol. Process., 6, 279-298, 1992.

Chandler, R., Lourmas, G. and Jesus, J.: MOMFIT Software for moment-based fitting of single-site stochastic rainfall model fitting, User guide, Department of Statistical Science, University College London, London, 2010.

Fawcett, L. and Walshaw, D.: Estimating return levels from serially dependent extremes, Environmetrics, 23, 272-283, 2012.

**1.2 Anonymous Referee #2**

**General comments**

The research work presented in the manuscript develops a new methodology to estimate fine-scale rainfall extremes. Although there has been a substantial amount of work done, by many authors over the years, on stochastic point process models for rainfall, most of the models proposed tend to underestimate the rainfall extremes at fine-scales. Estimation or reproduction of extreme rainfall at hourly and sub-hourly scales is a well-known problem. In this context, this paper attempts to address this problem by using a censored approach to model rainfall extremes. This is in a way similar to the Excess Over Threshold (EOT) method commonly used in extreme value modelling, but here a stochastic mechanistic model is used along with this idea. Application of this

novel idea of censored modelling approach is illustrated in the estimation of fine-scale rainfall extremes from two different regions to provide an improved representation of extremes.

250 The paper gives an excellent coverage of the history of work carried out in this area to convey the rationale for the need to study or explore alternative methods for fine-scale extremes.

The success of this new approach, of course depends heavily on the choice of the censor level and, hence, emphasis was placed on finding appropriate value of the censor for the application. If the estimation of extreme
255 rainfall is the main objective of the study then using this censored approach is certainly a useful tool and worthwhile addition to the existing methods.

One drawback in the proposed approach might be the amount of fine-tuning required to get the best set of potential estimates for the extreme rainfall with respect to model, its parameterisation, censor, statistics used in
260 fitting as well as aggregation levels. This level of tweaking or fine-tuning might prove to be a lot to generate sufficient interest amongst practitioners. The rationale behind the need to make these choices, however, has been explained in the manuscript though.

**Specific comments.**
265

**Line 257:** Would be useful to give a reason for the assumption of rain cells starting at the storm origin.

In the Bartlett-Lewis rectangular pulse (BL) models, it is assumed that rain cells start at the storm origin largely for mathematical convenience. Because the BL cluster mechanism is defined by the interval between successive
270 cells, a starting point is required. Therefore, it is convenient to assume that rain cells start at the storm origin. In contrast, the Neyman-Scott rectangular pulse (NS) cluster process is defined by the temporal distance between storm and cell origins and typically assumes that rain cells do not start at the storm origin. Again, this is largely for mathematical convenience.

275 In the case of the BL models, the assumption of rain cells starting at the storm origin prevents the simulation of empty storms which can occur if the first rain cell starts after the end of the storm. This issue does not arise in NS models because the number of cells per storm is a model parameter to be fitted, therefore a minimum value of one can be specified thus preventing the simulation of empty storms.

280 **Line 316-319:** Can appreciate the reason given for the choice of fitting statistics used for model calibration, but the question now is that how do the parameter estimates compare when the same fitting stats are used for uncensored fitting? Has this been explored?

This is an interesting question but one that we haven't explored. We will look into this although we don't feel that
285 it will change the analysis presented in this paper.

**Line 358:** Perhaps you need to explain what you meant by behavioural parameters for the readers.

This was also highlighted by Anonymous Referee #1 therefore the response below is the same as that provided
290 to referee #1.

We have used the term "behavioural parameters" by analogy with Beven and Binley (1992). We have used the term to refer to well identified models. We have found that for well identified parameters with narrow 95% confidence intervals, simulation bands on the extreme value estimates are correspondingly narrow. As the

parameters become less well identified, their 95% confidence intervals increase giving rise to extreme value estimates which deviate significantly from the observations, which in turn results in significant deviation of the simulation band upper limit. This effect is shown in Fig.11 resulting from the very large parameter uncertainty shown in Fig.12.

We will remove the reference to "behavioural parameterizations" in the context of this research and change all references to well identified parameters.

**Line 358: 95% confidence intervals:** unless you are using the standard errors of the estimates, I am not sure whether "confidence" interval is the appropriate terminology here. Simulation bands?

This was also highlighted by Anonymous Referee #1 therefore the response below is the same as that provided to referee #1. We agree with the suggestion and will change all occurrences in the manuscript.

There are in fact two issues here: if we were doing very long simulations with practically no random noise (so that another simulation would yield practically the same result), then we would have identified approximate confidence intervals. But with the shorter simulation length, both parameter uncertainty and the randomness of the model are combined in the spread we observe in the simulated statistics, so that 'simulation bands' is indeed a better descriptor.

**Line 396:** Not sure why your validation for Atherstone was based on 0.6mm censor which seem to contradict your statement on lines 375-380. Some insight/explanation would be useful to the readers.

We agree that it would have been more consistent to select 0.2 mm for validation of the Atherstone hourly censored model. We will change the selection in Table.2 to include 0.2 mm for the hourly resolution at Atherstone (as below). We will then revise the validation plots to suit.

**Table 2 Censor selection for model validation.**

|  | 5 minutes | 15 minutes | 60 minutes |
|---|---|---|---|
| Bochum | 0.5 mm | 1.0 mm | 1.0 mm |
| Atherstone | 0.6 mm | 0.6 mm | 0.2 mm |

**Line 396: Table 2.** Different censor for different sites is understandable. However, why do you need to use the same sensor at 3 different aggregation level for Atherstone while using different censors for the 3 levels of aggregation for Bochum?

The model is fitted separately for each temporal resolution which explains why we have different censors. Because of the effect of aggregation, we cannot use a model censored at one resolution to estimate rainfall extremes at a coarser one. Therefore, censored model parameters are scale dependent which is explained in section 3.

The censors given in Table.2 were chosen for validation. Our analyses show that there are a range of censors that could be applied giving improved estimation of extremes. In the case of the two sites investigated, the gauge resolution at Atherstone is much coarser than that at Bochum. We note on lines 504-7 (page 28) that that a censor of 0.5 mm for 15 minute rainfall at Atherstone gives very similar extreme value estimation to the selected 0.6 mm censor, implying that it may be sufficient at this site to limit the censor to the gauge resolution. At Bochum, the finer gauge resolution will capture rainfall amounts with greater accuracy than at Atherstone. Therefore, we

expect that there is greater capacity for the Bochum models to give improved estimation of extremes with increasing censors hence the different censors selected in Table.2.

**Fig 8: row 2.** The nice seasonal pattern observed in the mean rainfall for Atherstone at 5 and 15 minutes has become less prominent or disappeared at 60 minutes. Can you comment on why? No observation or comment was made about this.

The plots in Fig.8 show the summary statistics for censored rainfall with different censors applied in each column. While the censors chosen for validation in Table.2 are the same (0.6 mm), their effect on model fitting is different because they are applied to each temporal scale. Hence, when we look at the mean monthly rainfall in validation, we are looking at the seasonal variation in the rainfall after censoring.

Without censoring, the seasonal variation in mean monthly rainfall will only change in magnitude between scales. For a constant censor between scales as shown in panels d, e and f, the seasonal variation in mean monthly rainfall will vary between scales because there is a higher proportion of low observations at short temporal scales removed by the censors. The greater prominence in seasonal variation shown in plots d and e indicates that the summer months (apprx. Apr - Oct) are more prone to short intense bursts of rain, and the winter months longer periods of low rainfall intensity. This is consistent with there being more convective rainfall in the summer, and stratiform rainfall in the winter.

**References**

Beven, K. and Binley, A.: The future of distributed models: Model calibration and uncertainty prediction, Hydrol. Process., 6, 279-298, 1992.

**1.3 Handling editor**
Received: 3 December 2017

**Line 7:** There is an inaccurate statement in the abstract: "In a warming climate, the moisture holding capacity of the atmosphere is greater which increases the potential for short duration high intensity storm events".

The simplest and obvious remedy is to delete this sentence, at no cost at all.

But if the authors wish to keep it, they should first change the incorrect phrase "moisture holding capacity of the atmosphere". I have also used the incorrect verb "hold" in a similar situation, and López-Arias (2012) correctly pointed out: "This kind of sentence is widely known to be the source of a very common misconception regarding the behaviour of water vapour and the role of air (which is none) in the process of reaching saturation (see also my Reply, Koutsoyiannis, 2012). In addition, if the authors want to keep the sentence, they should also modify the remaining part which looks fuzzy (e.g., what does "potential" mean? why "short durations"?)

We are very grateful to the editor for bringing this inaccuracy to our attention, and completely agree with the explanation provided that the water vapour content of the atmosphere is not a function of the air in which it is a constituent part. The purpose of our original statement was to highlight one of the primary challenges in reliable extreme rainfall estimation necessary for long term planning and design. However, because we do not take into account a changing climate in this work we agree that the simplest and most obvious remedy is to delete this sentence.

**Line 17:** Second, I find problematic the first sentence in the Introduction: "With growing evidence that the frequency and intensity of short duration rainfall extremes are increasing with climate change (Stocker et al. 2013, Westra et al. 2014, Kendon et al. 2014), the need for reliable extreme value estimation techniques is becoming more pressing." Perhaps there is such growing evidence, perhaps not, but I do not think the references given are

sufficient: one is of political /policy orientation, one looks to refer to the future (can "evidence" be about future?) and one looks to be based on models.

Again, the simplest and obvious remedy is to delete this sentence, at no cost at all.

But if the authors wish to keep the statement, they need to rectify it, make it more scientific, and cite works using real world data: As a starting point, they may wish to cite Sun et al. (2012; excerpt "Here we analyse observations of monthly P (1940–2009) over the global land surface … We report a near-zero temporal trend in global mean P. Unexpectedly we found a reduction in global land P variance"); Stephens et al. (2010, notice the title of, "Dreary state of precipitation in global models"); and perhaps my own multirejected paper (Tsaknias et al., 2016)

*Upper limits on censoring were identified when model parameterization noticeably deteriorated resulting in the mean of the MVN realisations to deviate away from the optimal. Results presented are limited to the 4 highest censors with well identified model parameters, together with 95% simulation bands. The simulation bands show the range of extreme value estimation between the 2.5 and 97.5 quantiles of the 100 MVN realisations for each simulated data point.*

**New page 19,** Line 408 onwards: We have also changed the text as follows in relation to the presentation of validation statistics as the simulation bands here are estimated in a similar way.

*The plots show the estimated summary statistics calculated using the optimum parameter estimates, together with 95% simulation bands obtained by randomly sampling 100 parameter sets from the multivariate normal distribution of model parameters, estimating the summary statistics under the model and displaying the range of estimates between the 2.5 and 97.5 quantiles.*

16. In response to AR#1, **page 17, line 379**:
**New page 18,** Line 386 onwards: We have changed the text as follows.

*The mean of the MVN realisations for the BL1M model at Atherstone with the 0.6 and 0.8 mm censors (see Fig.6) diverges from the optimum because of the generation of unrealistic extremes.*

17. In response to AR#1, **pages 20-21, Sections 6.2.1, 6.2.2 and 6.2.3**:
**New page 18-21, Sections 6.2.1, 6.2.2 and 6.2.3**: We have made the following changes.

1. We have removed calibration plots for Bochum, but retained those for Atherstone.
2. We have updated the hourly plots for Atherstone to reflect the changed censor selection in Table 2.
3. We have removed the validation plots for the coefficient of skewness at both sited, but retained the plots for proportion of wet periods at both sites.
4. We have removed Section 6.2.3.

18. In response to AR#1, **page 27, Figure 14**:
**New page 26,** Fig.13: We have retained this figure and renumbered it following the removal of the profile objective function plots.

19. In response to AR#1, **page 28, line 514**:
**New page 26,** Line 514 onwards: We have introduced the rule to create independent peaks before Table 3. We have also removed the quantiles of the rainfall amounts from Table.3 as they are provided in Fig.13. The following change to the text is made.

*While the proportion of rainfall observations removed prior to model fitting is large - over 90% and 80% for 5 and 15 minute rainfall from Bochum and Atherstone respectively - comparison with the maximum rainfall amounts and an assessment of the number of independent peaks over the censor demonstrate that the censors are low. Table 3 shows the proportion of maximum rainfall and the number of independent peaks per year for the selected censors given in Table. 2. The number of peaks over the censors are estimated using a temporal separation of 48 hours to define independence.*

20. In response to AR#1, **pages 28-30**:
**New page 27-9,** Section 8: The manuscript has been updated as follows:

1. Section 8 has been renamed "Further discussion and conclusions"
2. Section 9 has been removed and combined with section 8.
3. The first sentence from the old Section 9 has been removed because it repeats the first sentence of Section 8. "Censored rainfall synthesis using mechanistic pulse based models appears to offer an alternative approach to estimation of rainfall extremes and to frequency estimation techniques"
4. The following changes have been made to the subsequent two sentences: *"The results presented in this paper show that the method has worked for single site data from two very different locations,*

*and recorded using different gauging techniques. Consistency in the relative magnitude of selected censors identified at each location, and the stability of estimation across a range of censors gives confidence in the approach and supports the original hypothesis."*

5. Discussion on the upward curvature in the Gumbel extreme value plots is included.

21. In response to AR#1, **page 31, line 591**:
**New page 30,** Line 608: The inequality is corrected.

22. In response to AR#1, **page 32, Figure A.2**:
**New page 31,** Fig.A2: The figure is corrected.

**New page 9,** Line 228: We have changed the text as follows.

*Both X and L are assumed to be independent of each other and follow exponential distributions with parameters $1/\mu_x$ and $\eta$ respectively.*

**2.2 In response to Anonymous Referee #2 (AR#2)**

1. In response to AR#2, **Line 257**:
**New page 10, lines 258 onwards:** We have update the manuscript with the following text.

*For the models used in this study, it is assumed that rain cells start at the storm origin to prevent the simulation of empty storms which can occur with the Bartlett-Lewis clustering mechanism if the first rain cell starts after the end of the storm.*

2. In response to AR#2, **Line 316-319**:
We have not update the manuscript with respect to this comment.

3. In response to AR#2, **Line 358**: (as per AR#1, comment no. 10)
The following changes have been made:

1. *Line 361 - Results presented are limited to the 4 highest censors with well identified model parameters, together with 95% confidence intervals derived from MVN realisations.*
2. *Line 446 - Figure 10 shows the change in 95% confidence intervals and the mean of the MVN realisations obtained with censored models with well and poorly identified parameters for 15 minute data at Bochum and Atherstone.*
3. *Line 465 - When sampling from the multivariate normal distribution for model parameters in simulation, these large uncertainties give rise to poor extreme value estimation.*
4. *Figure 10 Change in 95% confidence intervals and mean of the MVN realisations for Bochum and Atherstone 15 minute data with well identified (> 1.0 mm and > 0.6 mm) and poorly identified (> 1.5 mm and > 1.0 mm) censored model parameters.*

4. In response to AR#2, **Line 358**: (as per AR#1, comment no. 15)
**New page 14,** Line 360 onwards: We have changed the text as follows. We have also changed references to simulation bands in Figs. 4-6 and their captions.

*Upper limits on censoring were identified when model parameterization noticeably deteriorated resulting in the mean of the MVN realisations to deviate away from the optimal. Results presented are limited to the 4 highest censors with well identified model parameters, together with 95% simulation bands. The simulation bands show the range of extreme value estimation between the 2.5 and 97.5 quantiles of the 100 MVN realisations for each simulated data point.*

**New page 19,** Line 408 onwards: We have also changed the text as follows in relation to the presentation of validation statistics as the simulation bands here are estimated in a similar way.

*The plots show the estimated summary statistics calculated using the optimum parameter estimates, together with 95% simulation bands obtained by randomly sampling 100 parameter sets from the multivariate normal distribution of model parameters, estimating the summary statistics under the model and displaying the range of estimates between the 2.5 and 97.5 quantiles.*

5. In response to AR#2, **Line 396**: (as per AR#1, comment no. 6)
**New page 18,** Line 395 onwards: We have changed the selected censor for validation of hourly extremes at Atherstone to 0.2 mm. This is reflected in Table 2, Fig. 8 and generally within the text.

6. In response to AR#2, **Line 396: Table 2:**
We have not update the manuscript with respect to this comment.

7. In response to AR#2, **Fig 8: row 2:**
Section 6.2 has substantially changed following comments from both reviewers. The following changes listed in response to AR#1 comment no. 17 are relevant to this comment by AR#2.

**New page 18-21, Sections 6.2.1, 6.2.2 and 6.2.3**: We have made the following changes.

1. We have removed calibration plots for Bochum, but retained those for Atherstone.
2. We have updated the hourly plots for Atherstone to reflect the changed censor selection in Table 2.
3. We have removed the validation plots for the coefficient of skewness at both sited, but retained the plots for proportion of wet periods at both sites.
4. We have removed Section 6.2.3.

**New page 20, lines 415 onwards:** We have update the manuscript with the following commentary as suggested.

*All models perform very well with respect to replicating the summary statistics used in fitting with the 95% simulation bands comfortably bracketing the observations. The estimated summary statistics are very close to the observed with all models performing equally well. The seasonal variation in mean monthly rainfall varies between scales because there is a higher proportion of low observations at short temporal scales removed by the censors. The greater prominence in seasonal variation shown in plots a and b indicates that the summer months (approx. Apr - Oct) are more prone to short intense bursts of rain, and the winter months longer periods of low rainfall intensity. This is consistent with there being more convective rainfall in the summer, and stratiform rainfall in the winter. The plots in Fig. 8 demonstrate that the models are able to reproduce the censored fitting statistics, confirming reliability of the process.*

**New page 21, lines 431 onwards:** We have update the manuscript with the following additional commentary on the ability of the models to reproduce the proportion of wet periods.

*The ability of the models to reproduce the proportion of wet periods is generally good, although there is a tendency for all models to overestimate this statistic at both sites. At the 5 minute resolution for Bochum, the 95% simulation bands comfortably bracket the observations between the months of May and October, although there is over-estimation in the other months and for all months at the 15 and 60 minute scales. At Atherstone, there is good representation of the proportion of wet periods at the 15 minute scale, although over-estimation at the 5 and 60 minute scales. Generally, there is very slightly better agreement in the summer months which, as highlighted in Sect. 6.2.1, may be more prone to short intense downpours at fine temporal scales. This suggests that the censored models may be more effective at simulating the heavier short duration rainfall characteristic of summer convective storms, than the longer duration low intensity rainfall characteristic of winter storms.*

**2.3 In response to the Handling Editor (HE)**

1. In response to HE#1, **Line 7**:
We have removed this sentence.

2.  In response to HE#2, **Line 17**:
We have removed this sentence.

**3. Track changes to the manuscript**

640

*Manuscript commences on the next page.*

[revised manuscript text omitted]

Bochum 15 minute extremes

(a) BL0      (b) BL1      (c) BL1M

obs. AM                    opt. AM [> 0 mm]
opt. AM [> 1 mm]           opt. AM [> 1.5 mm]
mvn. AM [> 1 mm]           mvn. AM [> 1.5 mm]       mvn. RL [> 1 mm]       mvn. RL [> 1.5 mm]
95% SBs [> 1 mm]           95% SBs [> 1.5 mm]

Un−censored fitting timescales [h]:     Mean {1} CoeffVar {0.25,6,24} Skew {0.25,6,24} lag−1 AC {0.25,6,24}
Censored fitting timescales [h]:        Mean {1} CoeffVar {0.25,6,24} lag−1 AC {0.25,6,24}

Atherstone 15 minute extremes

(d) BL0      (e) BL1      (f) BL1M

obs. AM                    opt. AM [> 0 mm]
opt. AM [> 0.6 mm]         opt. AM [> 1 mm]
mvn. AM [> 0.6 mm]         mvn. AM [> 1 mm]         mvn. RL [> 0.6 mm]     mvn. RL [> 1 mm]
95% SBs [> 0.6 mm]         95% SBs [> 1 mm]

Un−censored fitting timescales [h]:  Mean {1} CoeffVar {0.25,6,24} Skew {0.25,1,6,12,24} lag−1 AC {0.25,1,6,12,24}
Censored fitting timescales [h]:     Mean {1} CoeffVar {0.25,6,24} lag−1 AC {0.25,6,24}

[revised manuscript text omitted]